# A single-cell survey of *Drosophila* blood

**Sudhir Gopal Tattikota[1]\*, Bumsik Cho[2†], Yifang Liu[1†], Yanhui Hu[1†], Victor Barrera[3], Michael J Steinbaugh[3], Sang-Ho Yoon[2], Aram Comjean[1], Fangge Li[1], Franz Dervis[1], Ruei-Jiun Hung[1], Jin-Wu Nam[2], Shannan Ho Sui[3], Jiwon Shim[2], Norbert Perrimon[1,4]\***

[1]Department of Genetics, Blavatnik Institute, Harvard Medical School, Boston, United States; [2]Department of Life Science, Hanyang University, Seoul, Republic of Korea; [3]Harvard TH Chan Bioinformatics Core, Boston, United States; [4]Howard Hughes Medical Institute, Boston, United States

**Abstract** *Drosophila* blood cells, called hemocytes, are classified into plasmatocytes, crystal cells, and lamellocytes based on the expression of a few marker genes and cell morphologies, which are inadequate to classify the complete hemocyte repertoire. Here, we used single-cell RNA sequencing (scRNA-seq) to map hemocytes across different inflammatory conditions in larvae. We resolved plasmatocytes into different states based on the expression of genes involved in cell cycle, antimicrobial response, and metabolism together with the identification of intermediate states. Further, we discovered rare subsets within crystal cells and lamellocytes that express fibroblast growth factor (FGF) ligand *branchless* and receptor *breathless*, respectively. We demonstrate that these FGF components are required for mediating effective immune responses against parasitoid wasp eggs, highlighting a novel role for FGF signaling in inter-hemocyte crosstalk. Our scRNA-seq analysis reveals the diversity of hemocytes and provides a rich resource of gene expression profiles for a systems-level understanding of their functions.

**\*For correspondence:**
sudhir_gt@hms.harvard.edu (SGT);
perrimon@receptor.med.harvard.edu (NP)

†These authors contributed equally to this work

## Introduction

The immune system forms an important layer of defense against pathogens in a wide variety of organisms including *Drosophila* (*Banerjee et al., 2019*; *Mathey-Prevot and Perrimon, 1998*). The chief mode of immune response in flies involves innate immunity, which is composed of diverse tissue types including fat body, gut, and blood cells called the hemocytes (*Buchon et al., 2014*). Hemocytes represent the myeloid-like immune cells, but so far have been considered less diverse compared to their vertebrate counterparts (*Evans et al., 2003*; *Wood and Martin, 2017*). In addition to progenitor cells or prohemocytes, three major types of hemocytes are known in *Drosophila*: plasmatocytes, crystal cells, and lamellocytes. Plasmatocytes are macrophage-like cells with parallels to vertebrate tissue macrophages, while crystal cells and lamellocytes perform functions analogous to clotting and granuloma formation in vertebrates (*Buchon et al., 2014*). Hemocytes in the larva derive from two lineages: the lymph gland and the embryonic lineage, which in the larva forms resident (sessile) clusters of hemocytes in subepidermal locations, also known as hematopoietic pockets (*Gold and Brückner, 2015*; *Holz et al., 2003*; *Jung et al., 2005*; *Lanot et al., 2001*; *Makhijani et al., 2011*). Prohemocytes can give rise to all mature hemocytes in the lymph gland (*Banerjee et al., 2019*). Likewise, the embryonic lineage, which consists of self-renewing plasmatocytes, is capable of producing crystal cells and lamellocytes during inflammatory conditions (*Gold and Brückner, 2015*; *Leitão and Sucena, 2015*; *Márkus et al., 2009*). Whereas plasmatocytes are important for phagocytosis and represent ~90–95% of total hemocytes, crystal cells, which constitute ~5%, are required for wound healing and melanization (*Banerjee et al., 2019*; *Evans et al., 2003*).

Traditionally, the classification of hemocytes is based on two major criteria: cell morphology (*Rizki, 1957*; *Rizki, 1962*; *Shrestha and Gateff, 1982*) and expression of a few marker genes (*Evans et al., 2003*; *Evans et al., 2014*; *Kurucz et al., 2007*). The paucity of markers available to define cell types and the low-resolution of cell morphologies may have hindered the identification of rare cell types and failed to distinguish transient states. For instance, a plasmatocyte-like cell called the podocyte, which possibly corresponds to an intermediate state between plasmatocytes and lamellocytes, has been reported but its transcriptional signature remains unknown (*Honti et al., 2010*; *Rizki, 1957*; *Stofanko et al., 2010*). Moreover, ultrastructural and microscopic evidence has also suggested that several subsets within plasmatocytes and crystal cells exist, but they have not been characterized at the molecular level (*Rizki, 1957*; *Shrestha and Gateff, 1982*). Finally, little is known about hemocyte lineage trajectories with regards to the source cells or precursors, or about intermediate states that exist on the path to terminal differentiation of mature cell types. Hence, it is important to thoroughly characterize the molecular signatures of all the dynamic states of mature cell types in steady state and inflammatory conditions.

Advances in single-cell RNA sequencing (scRNA-seq) technologies allow comprehensive characterization of complex tissues, including blood (*Satija and Shalek, 2014*). In particular, scRNA-seq is powerful not only for identifying cell types but also resolving cell states and their dynamic gene expression patterns that are often buried in bulk RNA measurements (*Trapnell, 2015*). For example, recent studies using various scRNA-seq platforms have helped identify novel subtypes within monocytes and dendritic cells (*Villani et al., 2017*) and activated states of T cells (*Szabo et al., 2019*) in human blood. Further, scRNA-seq has documented the continuous spectrum of differentiation along the hematopoietic lineage in various species (*Macaulay et al., 2016*; *Nestorowa et al., 2016*; *Velten et al., 2017*; *Zhang et al., 2018*). In addition, scRNA-seq data allows pseudotemporal ordering of cells to re-draw developmental trajectories of cellular lineages (*Cao et al., 2019*). Thus, scRNA-seq, in conjunction with various lineage trajectory algorithms, allows the precise characterization of 1. differentiated cell types and their subtypes, 2. transient intermediate states, 3. progenitor or precursor states, and 4. activated states, which are often influenced by mitotic, metabolic, or immune-activated gene modules (*Adlung and Amit, 2018*; *Trapnell, 2015*; *Wagner et al., 2016*).

Here, we performed scRNA-seq of *Drosophila* hemocytes in unwounded, wounded, and parasitic wasp infested larvae to comprehensively distinguish mature cell types from their transient intermediate states. Our scRNA-seq analysis identifies novel marker genes to existing cell types and distinguishes activated states within plasmatocytes enriched in various genes involved in the regulation of cell cycle, metabolism, and antimicrobial response. In addition, we could precisely distinguish mature crystal cells and lamellocytes from their respective intermediate states. Interestingly, our scRNA-seq revealed the expression of fibroblast growth factor (FGF) receptor *breathless* (*btl*) and its ligand *branchless* (*bnl*), in rare subsets of lamellocytes and crystal cells, respectively, which we implicate in regulating effective immune responses against parasitoid wasp eggs in vivo. Altogether, our scRNA-seq analysis documents the diversity of hemocyte cell populations circulating in the fly blood and provides a resource of gene expression profiles of the various cell types and their states in *Drosophila*.

## Results

### scRNA-seq of *Drosophila* hemocytes

Hemocyte differentiation can be induced in *Drosophila* larvae by mechanical wounding or oviposition by wasps such as *Leptopilina boulardi* (*Márkus et al., 2005*; *Rizki and Rizki, 1992*). Hence, to characterize hemocyte populations and their heterogeneity, we first performed the two immune responsive conditions: wounded and wasp 24 hr post-infested (wasp inf. 24 hr), together with unwounded control conditions (*Figure 1A*). Further, to mobilize the sessile hemocytes into circulation, we briefly vortexed the larvae prior to bleeding (*Petraki et al., 2015*). Subsequently, single hemocytes were encapsulated using microfluidics-based scRNA-seq technologies including inDrops (*Klein et al., 2015*), 10X Chromium (*Zheng et al., 2017*) or Drop-seq (*Macosko et al., 2015*). A total of 19,458 cells were profiled, with 3–4 replicates per condition, and obtained a median of 1010 genes and 2883 unique molecular identifiers (UMIs) per cell across all conditions (*Supplementary file 1*; *Figure 1—figure supplement 1A,B*). In order to achieve a comprehensive

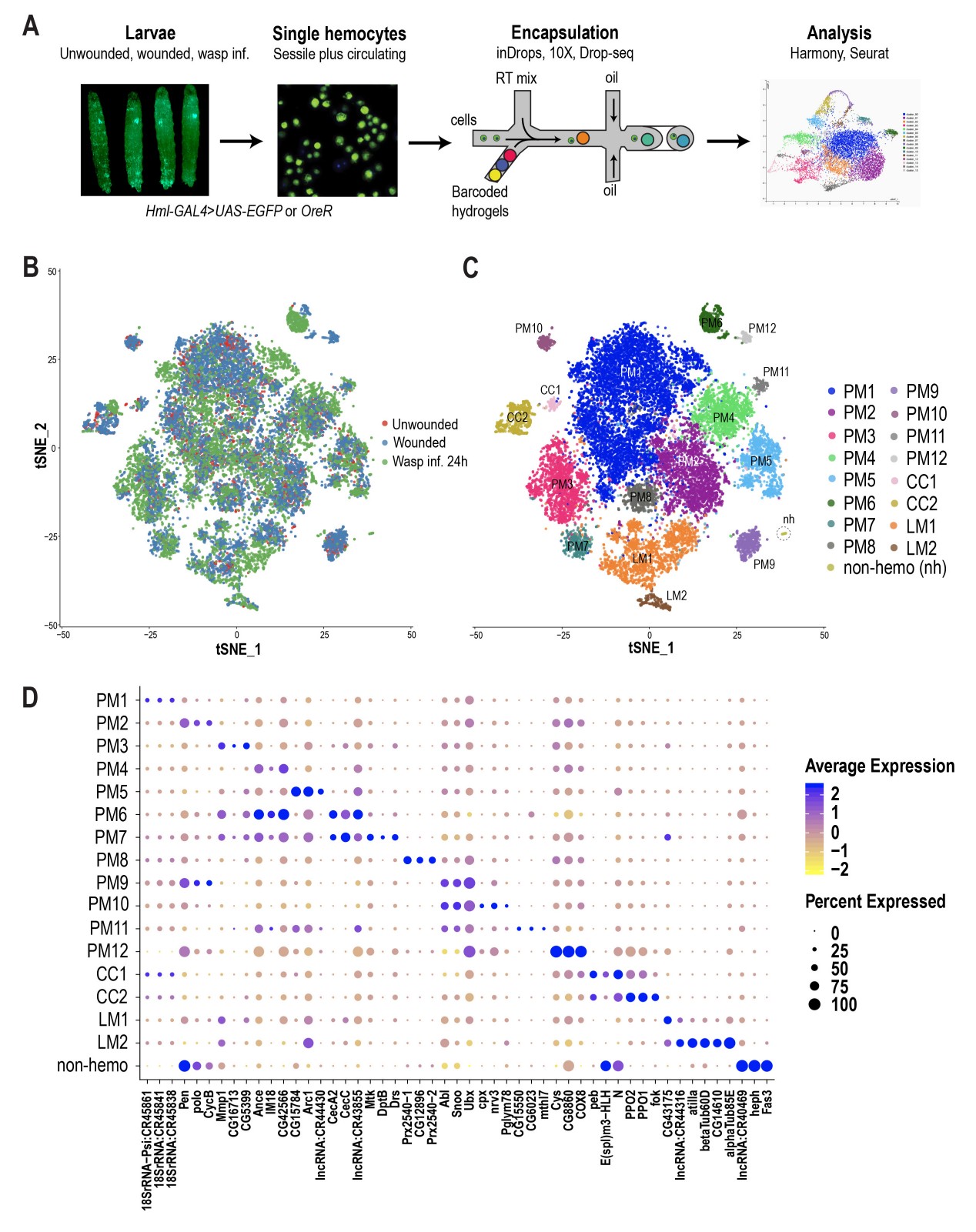

**Figure 1.** scRNA-seq of *Drosophila* hemocytes reveals subpopulations of plasmatocytes, crystal cells, and lamellocytes. (**A**) Schematic of the microfluidics-based scRNA-seq workflow. (**B**) t-Distributed Stochastic Neighbor Embedding (t-SNE) plot of Harmony-based batch correction and integration of unwounded (red), wounded (blue), and wasp inf. 24 hr (green) data sets. (**C**) Clustering of batch corrected cells from all three conditions

*Figure 1 continued on next page*

*Figure 1 continued*

reveals a total of 17 clusters. (D) Dot plot representing the top three genes enriched per cluster based on average expression (avg_logFC). Color gradient of the dot represents the expression level, while the size represents percentage of cells expressing any gene per cluster.

The online version of this article includes the following figure supplement(s) for figure 1:

**Figure supplement 1.** scRNA-seq of *Drosophila* hemocytes reveals subpopulations of plasmatocytes, crystal cells, and lamellocytes.
**Figure supplement 2.** Quality of scRNA-seq data and development of *Drosophila* blood scRNA-seq portal.

map of all the hemocytes profiled by the three scRNA-seq platforms, we merged all data sets. We observed notable 'batch effects' where cell types were being clustered according to condition, replicate, or technology (*Figure 1—figure supplement 1C,D,E*). Thus, we applied the Harmony batch correcting method (*Korsunsky et al., 2019*), which is integrated into the Seurat R package (*Stuart et al., 2019*). Harmony successfully integrated all three data sets, including their replicates (*Figure 1B*; *Figure 1—figure supplement 1D', E'*), and identified a total of 17 clusters (*Figure 1C*). Based on known marker genes, we confidently assigned certain clusters as plasmatocytes (marked by *NimC1*), crystal cells (marked by *lozenge* [*lz*]), or lamellocytes (marked by *atilla*) (*Figure 1C*; *Figure 1—figure supplement 1H*). Of the 17 clusters, one small cluster, representing ~0.2% of the total profiled cells, did not express any of the pan-hemocyte markers such as *Hemese* (*He*) and *serpent* (*srp*) (*Figure 1—figure supplement 1H*; *Evans et al., 2014*). Hence, we labeled this cluster as non-hemocyte (non-hemo).

## Diversity of hemocyte populations and their transcriptional dynamics
### Plasmatocyte clusters

Despite the fact that plasmatocytes constitute over 90–95% of the total hemocyte pool, subclasses within this major cell type have not been described. The majority of the clusters we identified (12/17) express *Hml* or *NimC1* but not crystal cell or lamellocyte markers and thus we annotated them as plasmatocyte (PM) clusters: PM1-12 (*Figure 1C*; *Figure 1—figure supplement 1H*).

PM1 represents the largest cluster and is enriched in rRNA genes such as *18SrRNA:CR45841* (*Figure 1D*). Interestingly, there is evidence in mammals that certain rRNA genes, including pre-rRNA molecules, accumulate in activated macrophages (*Radzioch et al., 1987*; *Varesio, 1985*). We determined PM2 as the cycling or self-renewing state of plasmatocytes based on the expression of genes related to cell cycle such as *CycB*, *stg*, and *polo*, which are markers of G2/M stages of the cell cycle (*Edgar and O'Farrell, 1990*; *Glover, 2005*; *Whitfield et al., 1990*). Next, we identified a group of PM clusters, PM3-5, which are enriched in several immune-induced genes, including *Matrix metalloproteinase 1* (*Mmp1*) and *Immune induced molecule 18* (*IM18*). Further, our scRNA-seq identified two distinct clusters, PM6 and 7, expressing several genes that encode antimicrobial peptides (AMP) (*Figure 1D*). PM6 highly expresses genes of the cecropin family, including *CecA2* and *CecC*, while PM7 is enriched in additional AMPs such as *Mtk*, *DptB*, and *Drs* (*Figure 1D*). The differential expression of this broad spectrum of AMPs (*Ferrandon et al., 1998*; *Hoffmann and Reichhart, 2002*; *Reichhart et al., 1992*; *Samakovlis et al., 1990*; *Tzou et al., 2002*) in PM6 and 7 (collectively termed PM$^{AMP}$) suggests that hemocytes can elicit a humoral immune response against a variety of pathogens. Together, we define PM3-7 as the immune-activated states of the plasmatocytes because of the expression of several immune-induced genes including the AMP-genes.

In addition to these major clusters, our scRNA-seq also identified several minor clusters: PM8-12. PM8 is enriched in genes encoding peroxidase enzymes such as *Prx2540-1*, *Prx2540-2*, and *CG12896*, which is an uncharacterized gene highly similar to *peroxiredoxin 6* (*PRDX6*) in humans. Although the function of these genes is not known in the context of hemocytes and immunity, peroxisomes have been reported to be necessary for phagocytosis by macrophages in both mice and *Drosophila* (*Di Cara et al., 2017*). PM9 and PM10 are enriched in the protooncogenes *Abl*, *Sno oncogene* (*Snoo*), and the transcriptional regulator *Ultrabithorax* (*Ubx*). PM11 and PM12 represent the smallest clusters within plasmatocytes and are defined by the expression of the uncharacterized genes *CG15550* and *CG6023*, together with a methuselah-type receptor gene, *methuselah-like 7* (*mthl7*) in PM11, and *Cys*, *CG8860*, and *COX8* in PM12 (*Figure 1D*; *Supplementary file 2*). In order to distinguish plasmatocytes in sessile or circulating compartments, we used quantitative real time PCR (qRT-PCR) of some of the marker genes per cluster in sessile and circulating hemocytes in

steady state unwounded larvae. Of all genes that we tested, only *Ubx* and *mthl7*, representing PM9-11, were relatively more enriched in sessile hemocytes compared to circulating hemocytes (*Figure 1—figure supplement 1G*), suggesting that most clusters may reside in both sessile and circulating compartments.

## Crystal cell clusters

The next most abundant immune responsive cells are the crystal cells, which are the main source of two enzymes important for melanization, prophenoloxidase 1 and 2 (PPO1 and PPO2). These enzymes are critical for survival upon wounding in larvae and adults (*Binggeli et al., 2014*; *Dudzic et al., 2019*; *Theopold et al., 2014*). Based on the expression of *PPO1* and *PPO2* together with the gene encoding the Runt related transcription factor *lozenge* (*lz*), two clusters were assigned to crystal cells (CC): CC1 and CC2 (*Figure 1C–D*; *Figure 1—figure supplement 1H*). CC1 expresses low levels of *PPO1* but high levels of *Notch*, *pebbled* (*peb*), and the enhancer of split complex gene *E(spl)m3-HLH* (*Figure 1D*). Although *Notch* and *peb* have been shown to be associated with crystal cell development (*Terriente-Felix et al., 2013*), expression of the Notch target gene *E(spl)m3-HLH* (*Couturier et al., 2019*) has not been reported. On the other hand, CC2 shows higher expression levels of *PPO1* and *PPO2* genes. Hence, we consider CC2 to represent mature crystal cells, while CC1 may represent an immature or a transient intermediate state.

## Lamellocyte clusters

Lamellocytes represent the rarest cell type, the numbers of which dramatically increase during wounding and wasp infestation (*Márkus et al., 2005*; *Rizki and Rizki, 1992*). Based on the expression of the lamellocyte marker gene *atilla* (*Evans et al., 2014*; *Kurucz et al., 2007*), we assigned two clusters to lamellocytes (LM): LM1 and LM2 (*Figure 1C–D*; *Figure 1—figure supplement 1J*). LM1 is the larger cluster of the two and is enriched in *atilla* besides a long non-coding RNA, *lncRNA: CR44316* (*Figure 1D*). LM2 represents a smaller cluster expressing *atilla*, *betaTub60D*, and *alphaTub85E*. The expression level of *atilla* is higher in LM2 compared to LM1 (*Figure 1D*), suggesting that LM2 represents mature lamellocytes, whereas LM1 may represent the lamellocyte intermediate state. Moreover, the strong expression pattern of tubulins and other cytoskeletal proteins may be important for the maintenance of structural integrity of lamellocytes and their dynamic roles in encapsulation (*Rizki and Rizki, 1994*).

Altogether, our scRNA-seq analysis recovered all major cell types within the hemocyte repertoire including the fine-grained dissection of plasmatocytes into self-renewing or cell-cycle (PM2) and various immune-activated states (PM3-7). The functions of the newly identified genes in the rest of the plasmatocyte clusters, which are minor subpopulations except for PM1, remain to be characterized. Presumably, these subpopulations represent transient intermediates along the course of terminal differentiation or activated states of plasmatocytes and other cell types. We also identified two clusters each for crystal cells and lamellocytes, which display differential expression of their marker genes, *PPO1* and *atilla*, respectively (*Figure 1D*; see *Supplementary file 2* for additional marker genes).

To validate our scRNA-seq data, we compared all the genes expressed in hemocytes of unwounded condition to publicly available bulk RNA-seq data sets of all larval hemocytes and crystal cells (*Miller et al., 2017*; *Neves et al., 2016*). To achieve this, we first converted our scRNA-seq data of all hemocytes and crystal cell clusters into pseudobulk RNA-seq and then compared with the published bulk RNA-seq data sets (*Miller et al., 2017*; *Neves et al., 2016*). We identified a strong correlation with spearman correlation of ~0.79 for both hemocyte and crystal cell comparisons, which reflects a high quality of our scRNA-seq data sets (*Figure 1—figure supplement 2A–B*). This resource can be mined using a user-friendly searchable web-tool (www.flyrnai.org/scRNA/blood/) where genes can be queried, visualized, and compared across conditions (*Figure 1—figure supplement 2C*).

## Changes in hemocyte composition and identification of a novel Mtk-like AMP

In addition to identifying genes enriched in each cluster and their changes across conditions, it is possible to estimate cell fraction changes from scRNA-seq data sets. To achieve this, we first segregated the three conditions (*Figure 2A–C*), then calculated cell fraction changes. Whereas PM1-5, 8,

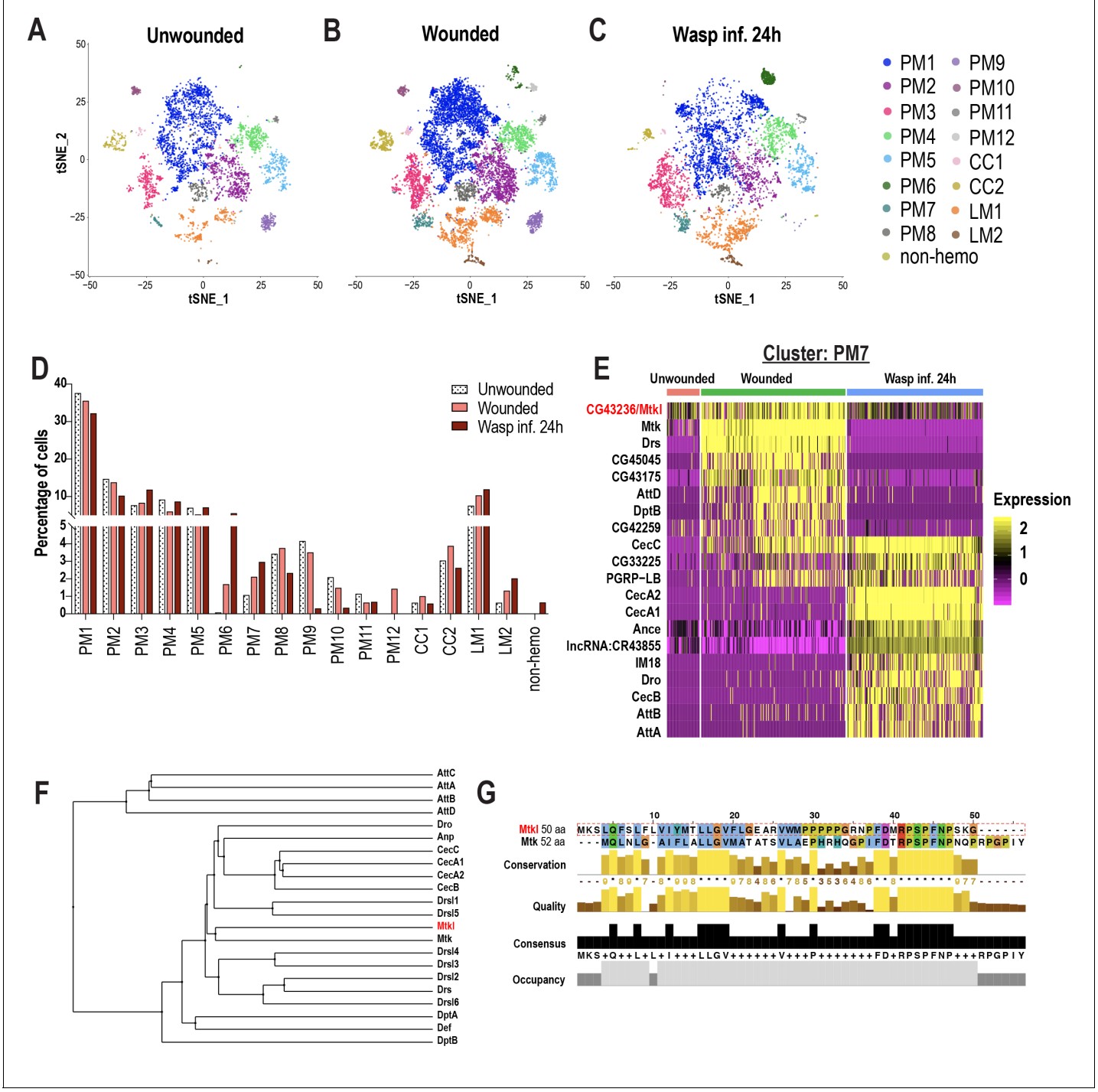

**Figure 2.** Changes in blood cell composition and identification of a novel Mtk-like AMP. (A-C) t-SNE plots of (A) Unwounded, (B) Wounded, and (C) Wasp inf. 24 hr conditions. (D) Cell fraction changes in clusters based on treatment conditions. (E) Heat map profile of the top expressed genes in cluster PM7 identifies *CG43236/Mtk-like* (*Mtkl*). Genes were ranked based on expression levels in each condition in the heat map. (F) Phylogenetic tree map constructed with the peptide sequences of all known AMPs together with Mtkl. (G) Global alignment of Mtkl and Mtk peptide sequences using Jalview protein alignment software (*Waterhouse et al., 2009*).

The online version of this article includes the following source data and figure supplement(s) for figure 2:

**Source data 1.** Source data pertaining to cell fraction bar graph of *Figure 2D*.

**Figure supplement 1.** Changes in blood cell composition and identification of a novel Mtk-like AMP.

and 11 are well represented in all three conditions, PM9 is negligibly detected in the wasp inf. 24 hr condition, and PM10 and 12 were majorly detected in the wounded condition. The PM^AMP clusters, PM6 and 7, emerged mainly upon wounding or wasp infestation compared to unwounded controls (*Figure 2A–D*). Strikingly, the proportion of *cecropin*-enriched PM6 cluster (~5.5%) is similar to the previously observed fraction of hemocytes (~5–10%) expressing *cecropin* genes upon infection (*Samakovlis et al., 1990*). With regard to the crystal cell clusters, both CC1 and CC2 were underrepresented in wasp inf. 24 hr, consistent with previous observations that crystal cell numbers dramatically decrease following oviposition by wasps (*Kacsoh and Schlenke, 2012*; *Figure 2A–D*). Of the lamellocyte clusters, LM1 was detected in all three conditions, however, LM2 emerged only upon wounding or wasp infestation. In summary, LM2 and PM^AMP are the major clusters that are represented mostly upon wounding or wasp infestation.

Despite some PM clusters being represented in all three conditions, their gene expression patterns are specific to wounding or wasp inf. 24 hr. For instance, *Mmp1* showed increased expression only upon wounding or wasp inf. 24 hr in PM3 compared to unwounded controls (*Figure 2—figure supplement 1A*). Likewise, the increased expression of GST genes such as *GstE6* is specific to wounded or wasp inf. 24 hr (*Figure 2—figure supplement 1B*). With regards to the PM6 cluster of immune-activated state, *CecA2* showed an increased expression specifically in wasp inf. 24 hr compared to wounded conditions (*Figure 2—figure supplement 1C*). Further, differentially expressed gene (DEG) analysis of PM7 revealed that most of the AMP genes were more enriched during wounding or wasp inf. 24 hr than in unwounded controls (*Figure 2E*). Of note, the AMP gene signature was unique to either wounded or wasp inf. 24 hr conditions. For example, whereas *Mtk*, *Drs*, *DptB*, and *AttD* were specific to wounded condition, *CecA1*, *CecB*, *Dro*, and *AttA* were unique to wasp inf. 24 hr (*Figure 2E*). These data indicate that clusters represented in all three conditions may nevertheless differ from each other in a condition-specific manner with respect to their differential gene signatures (*Supplementary file 3*).

A survey of all top enriched genes in PM7 revealed the identification of an uncharacterized gene, *CG43236*, which is relatively more enriched in wounded compared to unwounded or wasp infested conditions (*Figure 2E*). *CG43236* encodes a small peptide of 50 amino acids (aa) and its phylogenetic alignment with all known AMPs revealed that it clustered with Mtk, an antibacterial and antifungal AMP (*Levashina et al., 1995*; *Levashina et al., 1998*; *Figure 2F,G*). Both Mtk and CG43236 possess an Antimicrobial10 domain that is unique to the Metchnikowin family (*Figure 2G*; *Figure 2—figure supplement 1D–D'*), leading us to name *CG43236* as *Mtk-like (Mtkl)*. To validate its expression within hemocytes and whole larvae, we performed qRT-PCR in unwounded control and wounded conditions. Consistent with our scRNA-seq data (*Figure 2E*), *Mtkl* is well expressed and more enriched in hemocytes upon wounding in larvae. However, the induction of *Mtkl* was modest, compared to the robust induction of *CecC*, *Drs*, and *Mtk* (*Figure 2—figure supplement 1E*), suggesting that *Mtkl* may be strongly regulated by other modes of immune challenges such as sepsis or fungal infection. However, its expression is induced upon wounding in whole larvae and in fat bodies of wasp infested larvae (*Figure 2—figure supplement 1F–G*). Interestingly, a study has shown through comparative transcriptomics that the expression of *Mtkl* is upregulated in adult whole flies that were subjected to various types of bacterial infections (*Troha et al., 2018*). Furthermore, *CG43236/Mtkl* has been described as a putative AMP (*Troha et al., 2018*). Altogether, the DEG analysis of PM7 identified a novel Mtk-like putative AMP.

## Pseudotemporal ordering of cells delineates hemocyte lineages

Plasmatocytes of the embryonic lineage reside in larval hematopoietic pockets and, over the course of third larval instar, increasingly enter the hemolymph to circulate in the open circulatory system (*Makhijani et al., 2011*). The generation of these plasmatocytes, initially by differentiation from embryonic progenitors, and later in the larva through self-renewal of differentiated plasmatocytes, is well established (*Makhijani et al., 2011*). In contrast, the development of terminally differentiated crystal cells and lamellocytes from the embryonic lineage has remained speculative, but includes models of transdifferentiation from plasmatocytes (*Anderl et al., 2016*; *Leitão and Sucena, 2015*; *Márkus et al., 2009*). In our wounding experiments, although the lymph gland responds to the inflammatory stimulus, we did not detect any rupture or histolysis of lymph gland (*Figure 3—figure supplement 1A–E*), suggesting that this tissue may not be the major source of terminally differentiated cells such as lamellocytes in circulation. Hence, it is important to address the immediate sources

of mature cell types in circulation. scRNA-seq data can be used to construct pseudotemporal relationships between individual cell transcriptomes and impute cell lineages derived from precursor cells (*Trapnell, 2015*; *Trapnell et al., 2014*). We took advantage of our observation that PM2 expresses cell cycle genes to construct lineage trees emerging from this cell-cycle state. To avoid using batch corrected cells, we chose 10X genomics-derived unwounded and wounded data set, which represents all the mature cell types including lamellocytes that emerge only upon wounding (*Figure 3—figure supplement 1F*; *Supplementary file 1*). We used Monocle3 (*Cao et al., 2019*) and assigned PM2 as the start point of the pseudotime intervals (*Figure 3A–B*; *Figure 3—figure supplement 1G–H*). Monocle3 data shows that three major lineages emerge from the start point (*Figure 3C–E*). Lineage1 terminates in crystal cell (CC) fate, and includes the two CC clusters, CC1 and CC2. As expected, CC1 precedes CC2, strongly supporting that CC1 represents an intermediate state (CC$^{int}$) (*Figure 3B*). Consistently, the expression of *PPO1* steadily increases from CC1 and reaches its peak level upon becoming mature crystal cells (*Figure 3—figure supplement 2A*). Lineage2 on the other hand terminates in a fate that include PM1 and other minor PM clusters (PM8-12) (*Figure 3B,D*). Lastly, Lineage3 leads towards the clusters of the immune-activated state together with mature lamellocytes (*Figure 3B,E*). As expected, LM1 precedes LM2, supporting our initial observation that LM1 corresponds to the intermediate state of lamellocytes (LM$^{int}$). This is further demonstrated by the expression of *atilla*, which steadily increases with low levels in LM1 and higher levels in LM2 (*Figure 3—figure supplement 2B*). Lineage3 also terminates with cells of the immune-activated states, PM7 and PM5 (*Figure 3B*), which express *Drs* (in PM7) and *GstE6* (in PM5) (*Figure 3—figure supplement 2C–D*). Interestingly, PM5 is the only state that is enriched for the term xenobiotics biodegradation, suggesting a role for PM5 in this process (*Figure 3—figure supplement 2F*).

Next, to gain deeper insights into the gene expression signatures along the pseudotime intervals, we analyzed the differentially expressed genes (DEG) between the source and Lineage1 or Lineage3. DEG analysis over pseudotime revealed four major clusters depending on the expression of marker genes at the beginning (pre-branch) and end of the pseudotime interval along Lineage1 and 3 (*Figure 3F*). Interestingly, the pre-branch is enriched with PM marker genes such as *eater* and *Hml*, suggesting that precursor cells are indeed the self-renewing plasmatocytes. Moreover, the expression of the cell cycle genes gradually decreases as the lineages progress towards 1 or 3 (*Figure 3F*), consistent with a relationship between cell cycle arrest and terminal differentiation observed in vertebrates and flies (*Buttitta and Edgar, 2007*; *Guo et al., 2016*; *Morse et al., 1997*; *Ruijtenberg and van den Heuvel, 2016*; *Soufi and Dalton, 2016*). To assess whether blocking the cell cycle promotes terminal differentiation, we expressed RNAi against one of the top enriched cell cycle genes, *polo*, in Hml$^+$ plasmatocytes (*Hml-GAL4 >polo$^{RNAi}$*). RNAi-mediated knockdown of *polo* resulted in a significant increase in the production of lamellocytes compared to controls (*Figure 3—figure supplement 2G–I*), suggesting that cell cycle arrest may be required for terminal differentiation of cell types. Finally, to visualize cells accumulating along the pseudotime intervals, we analyzed the cell densities using ridge plots, which revealed that PM8-12 accumulated predominantly between the start and end of pseudotime intervals, indicating that they are all transient plasmatocyte intermediates (PM$^{int}$) (*Figure 3—figure supplement 2E*).

Altogether, based on Monocle3, we propose that PM2 has oligopotent potential and can give rise to terminally differentiated cell types and possibly other activated states within plasmatocytes. Further, our analysis confirms the existence of crystal cell and lamellocyte intermediate states that precede their fully differentiated mature cell types (*Figure 3G*).

## Crystal cell sub-clustering distinguishes crystal cell intermediates from mature crystal cells

Because crystal cells split into two distinct clusters (CC1 and CC2) (*Figure 1C*), we sub-clustered these cells independently of the other clusters. We used Harmony to correct for batch effects arising from the technological platforms and conditions (*Figure 4—figure supplement 1A–C*). Subsequent cell clustering revealed two distinct clusters: one with low *PPO1* expression (PPO1$^{low}$) and one with very high *PPO1* expression (PPO1$^{high}$) (*Figure 4A,C*). While the percentage of PPO1$^{low}$ crystal cells increased, the mature PPO1$^{high}$ crystal cell population shows a decreased trend upon wounding or wasp infestation (*Figure 4B*, *Figure 4—figure supplement 1D–F*). Furthermore, the expression level of *PPO1* was similar between unwounded controls and wounded condition in both clusters.

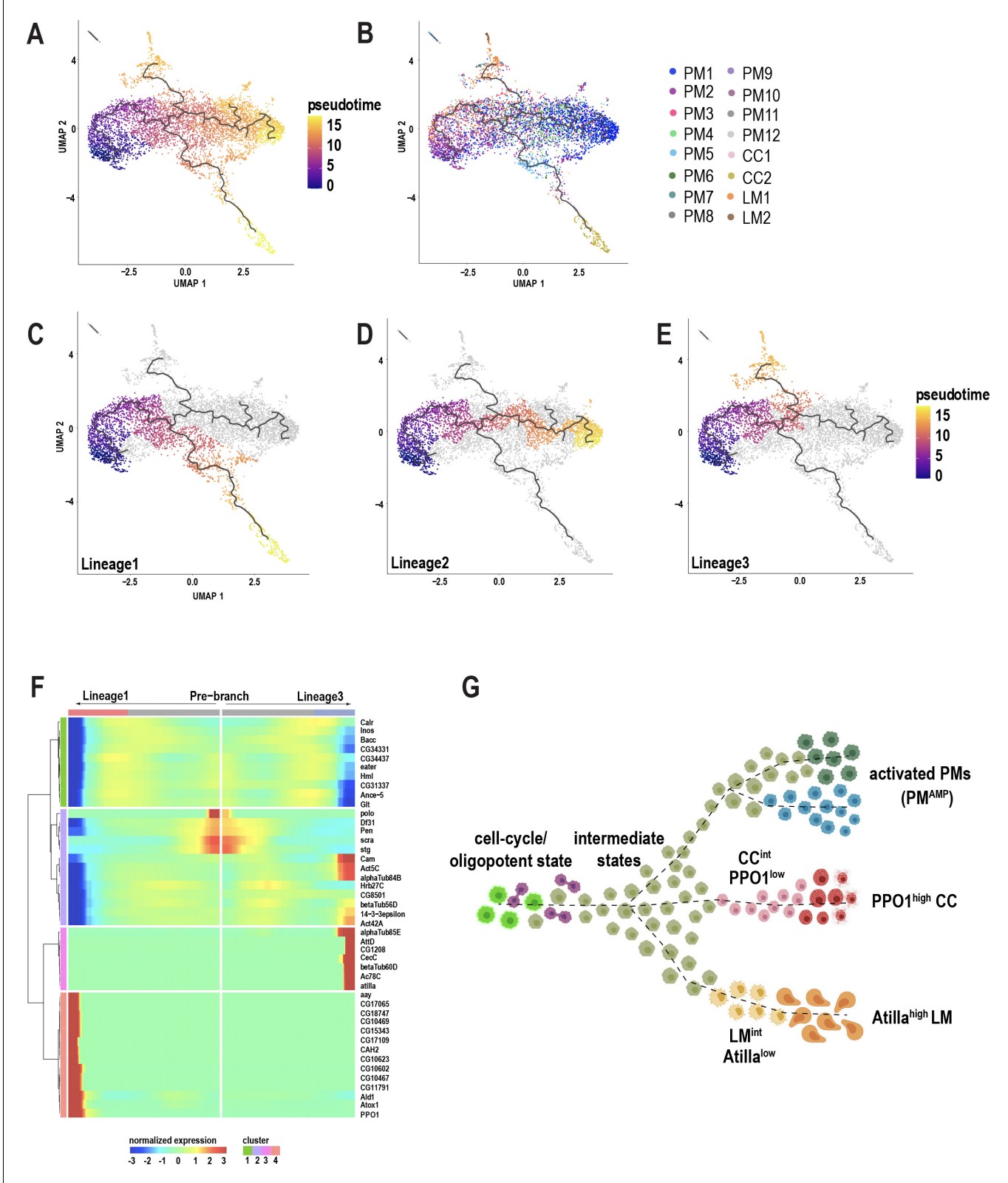

**Figure 3.** Pseudotemporal ordering of cells using Monocle3 delineates blood cell lineages. (**A**) Monocle3 was used to track cells over pseudotime on the 10X-derived unwounded and wounded data sets. (**B**) Visualization of clusters (from *Figure 1C*) onto the pseudotime map. (**C-E**) Three major lineage routes were obtained from the start site: Lineage 1 (**C**), Lineage 2 (**D**), and Lineage 3 (**E**). (**F**) Monocle-based gene expression signature between

*Figure 3 continued*

Lineages 1 and 3 with the 'pre-branch' in the middle. (**G**) Schematic showing potential lineage flow from the oligopotent state of plasmatocytes (PM2) to mature cell types with their intermediates.

The online version of this article includes the following figure supplement(s) for figure 3:

**Figure supplement 1.** Pseudotemporal ordering of cells using Monocle3 delineates blood cell lineages.

**Figure supplement 2.** Pseudotemporal ordering of cells using Monocle3 delineates blood cell lineages.

Interestingly, *PPO1* expression was negligibly detected in PPO1$^{low}$crystal cells and slightly lower in PPO1$^{high}$crystal cells in wasp infested larvae (*Figure 4D*). To determine whether the two crystal cell clusters coexist as distinct populations or whether PPO1$^{low}$ is a CC$^{int}$ state along the course of crystal cell maturation, we used *lz-GAL4, UAS-GFP; BcF6-mCherry* larvae to label Lz$^+$ and PPO1$^+$-crystal cells with GFP and mCherry, respectively. We examined the dorso-posterior end, where clusters of hemocytes that include crystal cells reside along the dorsal vessel of third instar larvae (*Figure 4E*; *Leitão and Sucena, 2015*). In line with the scRNA-seq data, in vivo imaging analysis revealed distinct populations of crystal cells within the sessile hub, with crystal cells displaying differential intensities of GFP and mCherry (*Figure 4F–F''*). Furthermore, intensity measurements of GFP and mCherry revealed a significant positive correlation (*Figure 4G*), which is consistent with previous studies demonstrating that Lz, together with Srp, can activate the expression of *PPO1* (*Waltzer et al., 2003*). Of note, the correlation plot did not reveal two separate populations of crystal cells, but rather heterogenous cell populations, existing in a potential continuum (*Figure 4G*). This further supports the Monocle3 prediction that PPO1$^{low}$ may represent the CC$^{int}$ state, whereas PPO1$^{high}$ corresponds to mature crystal cells (*Figure 3B*; *Figure 3—figure supplement 2A*).

As noted above, PPO1$^{high}$crystal cells express many of the mature crystal cell marker genes, including *PPO2* (*Figure 4H*). This cluster also highly expresses a number of uncharacterized genes such as *CG10602* and *CG10467*, which potentially encode enzymes with epoxide hydrolase and aldose 1-epimerase activities, respectively (*Figure 4H*). The human ortholog of *CG10602*, *LTA4H* (*leukotriene A4 hydrolase*), encodes an enzyme involved in the biosynthesis of a proinflammatory mediator, leukotriene B4 (*Crooks and Stockley, 1998*). Moreover, mutations in *lta4h* render zebrafish hypersusceptible to mycobacterial infections (*Tobin et al., 2010*). These observations suggest that *CG10602* may play an important role for mature crystal cells in combating bacterial infections, in addition to their role in melanization. On the contrary, PPO1$^{low}$crystal cells are enriched in *spatzle* (*spz*), a cytokine that activates the Toll pathway (*Lemaitre et al., 1996*; *Figure 4H*). Besides *spz*, PPO1$^{low}$crystal cells express cell cycle/chromatin associated genes such as the *Decondensation factor 31* (*Df31*) and *HmgD* (*Figure 4H*), suggesting that PPO1$^{low}$ may be in a cycling or proliferative state. Furthermore, although many of the genes, including *spz*, *Df31*, and *HmgD,* were more enriched upon wasp inf. 24 hr in PPO1$^{low}$ (*Figure 4I*; *Supplementary file 4*), the PPO1$^{high}$ cluster did not display notable changes in gene expression (*Figure 4—figure supplement 1G*). Of note, we confirmed the expression of a novel crystal cell enriched marker gene *E(spl)m3-HLH*, which is expressed in ~13% of crystal cells in both PPO1$^{low}$ and PPO1$^{high}$ populations (*Figure 4—figure supplement 1H–H'''*). In summary, crystal cell sub-clustering identifies a crystal cell intermediate state and highlights the possibility that crystal cells exist in a continuum, along the course of crystal cell maturation, as suggested by Monocle3 and in vivo imaging data.

## Lamellocyte sub-clustering identifies lamellocyte intermediates and subtypes

Previous studies have speculated the presence of lamellocyte intermediates (*Anderl et al., 2016*; *Honti et al., 2010*; *Rizki, 1957*; *Stofanko et al., 2010*). Monocle3 predicted that LM1 might correspond to a lamellocyte intermediate state (*Figure 3B,E*). To further test this prediction, we sub-clustered all the Atilla$^+$ lamellocytes derived from all conditions. To increase the diversity of lamellocytes, we performed scRNA-seq at one additional time point of wasp infestation: wasp inf. 48 hr. Clustering analysis revealed that more than 50% of all cells are *atilla*$^+$ lamellocytes (*Figure 5—figure supplement 1A–C*). To sub-cluster the lamellocytes, we considered the Atilla$^+$ clusters 0 and 1 from wasp inf. 48 hr (*Figure 5—figure supplement 1A–B*) together with Atilla$^+$ lamellocytes from unwounded, wounded, and wasp inf. 24 hr data sets (*Figure 1C–D*). We used Harmony to correct

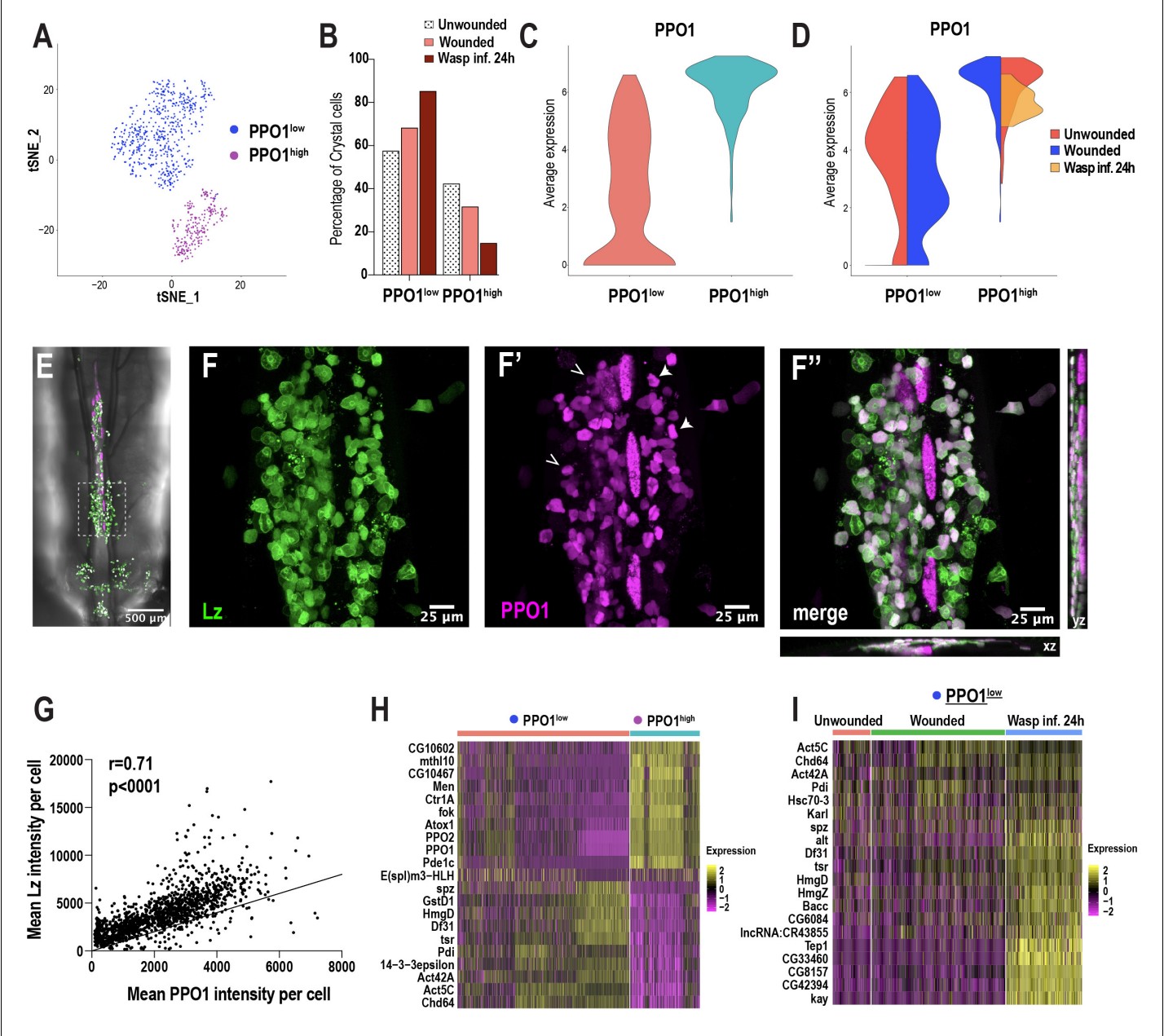

**Figure 4.** Crystal cell sub-clustering distinguishes crystal cell intermediates from mature crystal cells. (**A**) t-SNE plot of crystal cell sub-clustering depicting two crystal cell sub-clusters, PPO1[low] and PPO1[high]. (**B**) Percentage of PPO1[low] and PPO1[high] crystal cells across the three conditions. (**C**) Violin plot indicating the average expression level of *PPO1* in the two crystal cell clusters. (**D**) Average expression level of *PPO1* across the three conditions. (**E**) Confocal image of the posterior-dorsal side of a representative *lz-GAL4; UAS-GFP, BcF6-mCherry* third instar larva. BcF6-mCherry is a reporter for PPO1[+] crystal cells. Scale bar = 500 μm. (**F-F''**) Confocal images of GFP+ (F), BcF6-mCherry+ (F'), and merged GFP+ mCherry+ crystal cells (F''). xz and yz images in F'' represent the depth of the stacks. Representative PPO1[low] and PPO1[high] crystal cells are shown by open and solid arrow heads, respectively, in F'. Scale bar = 25 μm. (**G**) Mean intensities of GFP (Lz) and mCherry (PPO1). The correlation plot represents data from unwounded *lz-GAL4; UAS-GFP, BcF6-mCherry* larvae (n = 23; total crystal cells analyzed = 1397). The Pearson's correlation coefficient (r) and the p value (two-tailed) were calculated using Prism 8. (**H-I**) Heat maps of marker gene expression in PPO1[low] and PPO1[high] clusters (H) and differentially expressed gene (DEG) analysis of the marker genes across conditions in PPO1[low] cluster (I).

The online version of this article includes the following source data and figure supplement(s) for figure 4:

**Source data 1.** Source data pertaining to cell fraction bar graph of *Figure 4B*.

**Source data 2.** Excel file for *Figure 4G* pertaining to raw intensity values of Lz+ PPO1+ crystal cells.

**Figure supplement 1.** Crystal cell sub-clustering distinguishes crystal cell intermediates from mature crystal cells.

for batch effects (*Figure 5—figure supplement 1D–E*) and subsequent clustering of all the lamellocytes revealed 5 distinct clusters (*Figure 5A*), that we named LM1-4 and CC based on the expression of top enriched genes (*Figure 5E*). Cell fraction calculations revealed a higher fraction of LM1 upon wounding or wasp infestation compared to unwounded controls. However, the lamellocyte sub-clusters LM2-4 emerged mostly in wounding or wasp infestation (*Figure 5B*; *Figure 5—figure supplement 1F–I*). The last cluster was annotated as CC based on extremely low or no expression of *atilla* and enrichment of crystal cell marker genes including *PPO1* (*Figure 5C,E*). We speculate that LM1 may represent a LM^int state based on the low level of *atilla* together with enrichment of plasmatocyte marker genes such as *Pxn* and *Hml*, which are usually not expressed upon lamellocyte maturation (*Figure 2I*; *Figure 5E*; *Stofanko et al., 2010*). However, the expression of *atilla* is negligible in unwounded control and wounded conditions, while *Hml* is higher compared to wasp inf. conditions in LM1 (*Figure 5D*; *Figure 5—figure supplement 1J*).

Based on the high level of expression of *atilla*, we conclude that LM2, LM3, and LM4 are subtypes of mature lamellocytes (*Figure 5C*). Whereas LM3 and LM4 expressed relatively low levels of *Trehalase* (*Treh*) and its transporter *Tret1-1*, LM2 displayed high levels of these genes (*Figure 5E*). In addition to *Treh* and *Tret1-1*, almost all LM2 cells express two uncharacterized genes that potentially encode sugar transporters, *CG4607* and *CG1208* (*Figure 5E,F*). These genes emerged as the top-expressed genes in wasp inf. 48 hr when compared to the unwounded control, wounded, or wasp inf. 24 hr conditions (*Figure 5F*; *Supplementary file 5*). Of note, CG4607 and CG1208 belong to the SLC2 family of hexose sugar transporters and are orthologous to SLC2A6/GLUT6 or SLC2A8/GLUT8 in humans. Importantly, it has recently been demonstrated that GLUT6 acts as a lysosomal transporter, which is regulated by inflammatory stimuli (*Maedera et al., 2019*) and is strongly upregulated in macrophages activated by LPS (*Caruana et al., 2019*). Similarly, in *Drosophila*, immune activation is energy demanding and glucose may be used as a major source of energy for the production of mature lamellocytes (*Bajgar et al., 2015*). These observations suggest that lamellocytes may require specialized sugar transporters for their maturation to elicit an effective immune response against parasitoid wasp eggs.

In addition to the expression of the sugar transporters, most lamellocyte sub-clusters are enriched in a water transporter, Drip (AQP4 in humans), which has recently been shown to play a role in T-cell proliferation and activation in mice (*Nicosia et al., 2019*; *Figure 5F*; *Figure 5—figure supplement 1J–L*). To confirm *Drip* expression in vivo, third instar *Drip-GAL4, UAS-mCD8-GFP; BcF6-mCherry* larvae were wounded and then monitored for the expression of *Drip*. As expected, Drip^+ GFP cells were not detected during steady state in unwounded control larvae (*Figure 5G–G''' and I*). However, Drip^+ GFP cells, significantly increased upon wounding, and strikingly, the expression of Drip was detected in ~90% of Atilla^+ lamellocytes (*Figure 5H–H''' and I*). Finally, the smaller clusters of lamellocytes, LM3 and LM4, were also enriched in genes similar to those expressed in LM2, albeit at a lower level (*Figure 5E*), suggesting that LM3-4 are subtypes of mature lamellocytes. In contrast to LM3, LM4 expresses additional marker genes that include *GstD3* suggesting that this subtype of lamellocytes may also be involved in the detoxification of xenobiotics (*Figure 5E*; *Figure 5—figure supplement 1L*). Our DEG analysis also reveal distinct transcriptional changes in all lamellocyte sub-clusters of wasp inf. 48 hr compared to wounded or wasp inf. 24 hr conditions (*Figure 5F*, *Figure 5—figure supplement 1J–L*; *Supplementary file 5*). These differences are likely due to the origins of lamellocytes that are derived from either circulating hemocytes or lymph glands, which are ruptured in wasp inf. 48 hr time point unlike the intact lymph glands in wounded and wasp inf. 24 hr larvae (*Figure 3—figure supplement 1A–C*; *Cho et al., 2020*). In summary, LM sub-clustering identifies the LM^int state together with novel marker genes specific to different subsets of mature lamellocytes across different conditions.

## scRNA-seq uncovers a novel role for the FGF pathway in immune response

To identify signaling pathways enriched in each cluster, we performed pathway enrichment analysis on the scRNA-seq data across all conditions (*Figure 6A*). As expected, the PM6 and PM7 clusters are enriched in Imd and to lesser extent Toll signaling pathways (*Lemaitre and Hoffmann, 2007*). CC1 and CC2 are highly enriched in Notch signaling components (*Figure 6A*; *Supplementary file 6*), which is consistent with previous studies that showed the importance of Notch in crystal cell development (*Duvic et al., 2002*; *Leitão and Sucena, 2015*). Further, PM3 is highly enriched in

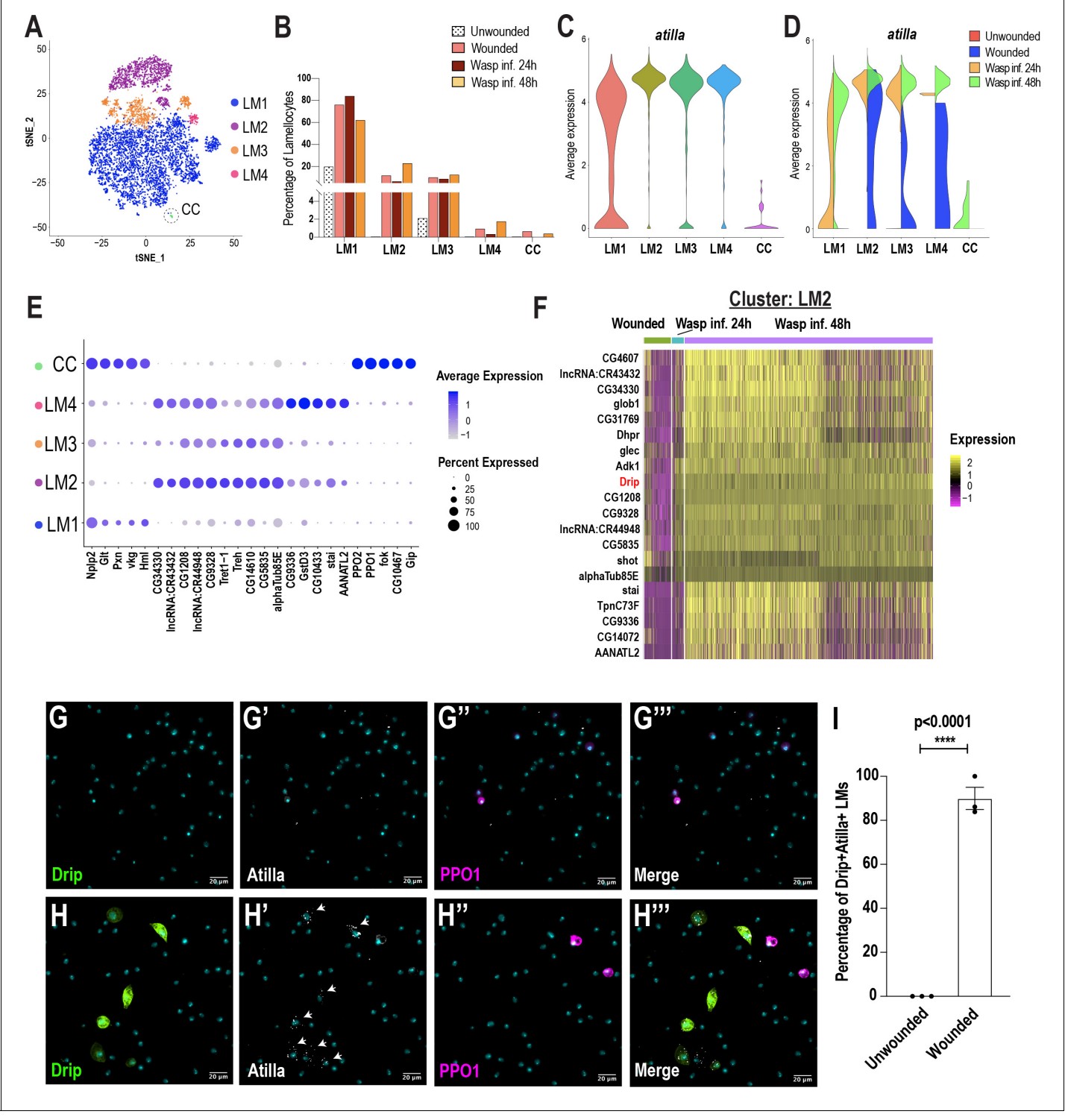

**Figure 5.** Lamellocyte sub-clustering identifies lamellocyte intermediates and subtypes. (A) t-SNE plot depicting the lamellocyte sub-clusters. (B) Changes in lamellocyte fractions across the three conditions. (C) Expression of the lamellocyte marker gene *atilla* was used to annotate the lamellocyte clusters. (D) Split violin plot shows the differential expression of *atilla* in different conditions. (E) Dot plot representing the top 5 marker genes per lamellocyte cluster. (F) Heat map depicts the DEG analysis of top genes in LM2 across all conditions. (G-H''') Expression validation of *Drip* in hemocytes derived from *Drip-GAL4 >mCD8 GFP; BcF6-mCherry* unwounded (n = 3 with 16 larvae per n) (G-G''') or wounded (n = 3 with 16 larvae per n) (H-H''') larvae. Nuclei are stained with DAPI (Cyan). Scale bar = 20 μm. (I) Percentage of GFP+ Atilla+ lamellocytes normalized to total Atilla+ lamellocytes per field of view. Data is represented by three independent biological replicates (n = 3). The error bars are represented as ± SEM (standard error of mean). The P value (unpaired t-test) is represented by **** (p<0.0001).

*Figure 5 continued on next page*

*Figure 5 continued*

The online version of this article includes the following source data and figure supplement(s) for figure 5:

**Source data 1.** Source data pertaining to cell fraction bar graph of *Figure 5B*.
**Source data 2.** Excel sheet pertaining to the lamellocyte counts used for *Figure 5I*.
**Figure supplement 1.** Lamellocyte sub-clustering identifies lamellocyte intermediates and subtypes.

particular for components of the TNF/Eiger, Imd, and Toll pathways. In addition to these known pathways, we also identified the fibroblast growth factor (FGF) signaling pathway to be highly enriched in the lamellocytes (*Figure 6A*; *Supplementary file 6*). Although the fly FGFR, Heartless (Htl) and its ligands, Thisbe (Ths) and Pyramus (Pyr) have been shown to be required for progenitor differentiation in the lymph gland (*Dragojlovic-Munther and Martinez-Agosto, 2013*), the role of FGF signaling has not been addressed in circulating hemocytes of the embryonic lineage. Interestingly, the second FGFR gene, *breathless* (*btl*), is one of the components that is enriched albeit at low levels in LM2 (*Figure 6B,B'*). In addition, a small subpopulation of crystal cells expresses *branchless* (*bnl*), which encodes the only ligand for Btl (*Figure 6B,B'*).

To confirm the expression of *bnl* in hemocytes upon wounding, we used *bnl-LexA; LexAOp-GFP, BcF6-mCherry* larvae and found that *bnl* expression was restricted either to crystal cells or plasmatocytes but not lamellocytes (*Figure 6C,C'*). In addition, based on *bnl* expression counts from scRNA-seq, *bnl* may be enriched in PPO1$^{high}$ compared to PPO1$^{low}$ crystal cells (*Figure 6—figure supplement 1A*). Similar to wounding, we also determined that *bnl* is expressed in subsets of crystal cells and plasmatocytes but not in lamellocytes 48 hr post infestation (PI) of *bnl-LexA; LexAOp-mCherry* larvae (*Figure 6D,D'*). The fraction of Bnl$^+$ crystal cells (as determined by mCherry$^+$ Hindsight/Hnt$^+$ cells [Hnt/Peb marks crystal cells]) was very low (~15.1% ± 17.97 standard deviation [SD]; n = 20) when compared to the total number of crystal cells. Next, we examined the expression of *btl* in hemocytes using *btl-GAL4; UAS-GFPN-lacZ, msn-mCherry*, where *msn-mCherry* marks lamellocytes following wasp infestation (*Tokusumi et al., 2009*). Confocal imaging at the vicinity of melanized wasp eggs in larvae revealed that a fraction of Btl$^+$ lamellocytes were detected near the melanized region, presumably encapsulating the wasp eggs (*Figure 6E–E'''*). Similar to the low fraction of Bnl$^+$ - crystal cells, the fraction of Btl$^+$ lamellocytes (as determined by GFP$^+$ msn$^+$ cells) is ~35% (±12.61 SD, n = 5) compared to the total lamellocytes per field of view (*Figure 6—figure supplement 1E–E'''*). In summary, these results confirm that both *bnl* and *btl* are expressed in rare subsets of crystal cells and lamellocytes, respectively. Besides their expression in circulation, we find that both *bnl* and *btl* are also expressed in lymph gland scRNA- and bulk RNA-seq data sets (*Figure 6—figure supplement 1C*; *Cho et al., 2020*). Importantly, *bnl* and *btl* are enriched in crystal cell and lamellocyte clusters, respectively, of the lymph gland scRNA-seq data (*Cho et al., 2020*; *Figure 6—figure supplement 1D*).

To characterize the role of *bnl* and *btl* in hemocytes, we used the wasp infestation model and knocked down *bnl* in most hemocytes using Hml-GAL4. Wasps were allowed to infest second instar *Hml-GAL4 >control* or *Hml-GAL4 >bnl$^{RNAi}$* (*bnl$^{RNAi}$*) larvae. 48 hr PI,~90% of control larvae (111/124) displayed melanotic nodules, which reflect the melanized wasp eggs. In stark contrast, the melanization frequency dropped to <10% (12/128) in *bnl$^{RNAi}$* larvae (*Figure 6H*), suggesting a defect in melanization. While crystal cell numbers were surprisingly unaltered, the number of all blood cells (DAPI$^+$ nuclei), Hml$^+$ plasmatocytes, and lamellocytes was significantly decreased in infested larvae (*Figure 6—figure supplement 1F–I*). On the other hand, total blood cell number including Hml$^+$ plasmatocytes, and crystal cells remained unaltered in uninfested control larvae (*Figure 6—figure supplement 1F–I*). Next, to address the role of *bnl* specifically in crystal cells, we used lz-GAL4 to drive RNAi against *bnl*. Similar to the results obtained with Hml-GAL4, over 80% of control larvae (84/99) showed melanized wasp eggs but the melanization frequency dropped to ~20% (31/166) in *lz-GAL4 >bnl$^{RNAi}$* larvae (*Figure 6F,H*), possibly due to the decline in the total blood cell number upon infestation (*Figure 6I–J*; *Figure 6—figure supplement 1J*). However, the number of crystal cells remained unchanged in infested control and *bnl$^{RNAi}$* larvae despite a decline in their numbers in uninfested condition in *bnl$^{RNAi}$* compared to controls (*Figure 6—figure supplement 1K*). Further, loss of *bnl* in crystal cells did not alter the number of Pxn$^+$ plasmatocytes in uninfested larvae (*Figure 6—figure supplement 1L*). Interestingly, we observed normal crystal cell morphology upon loss

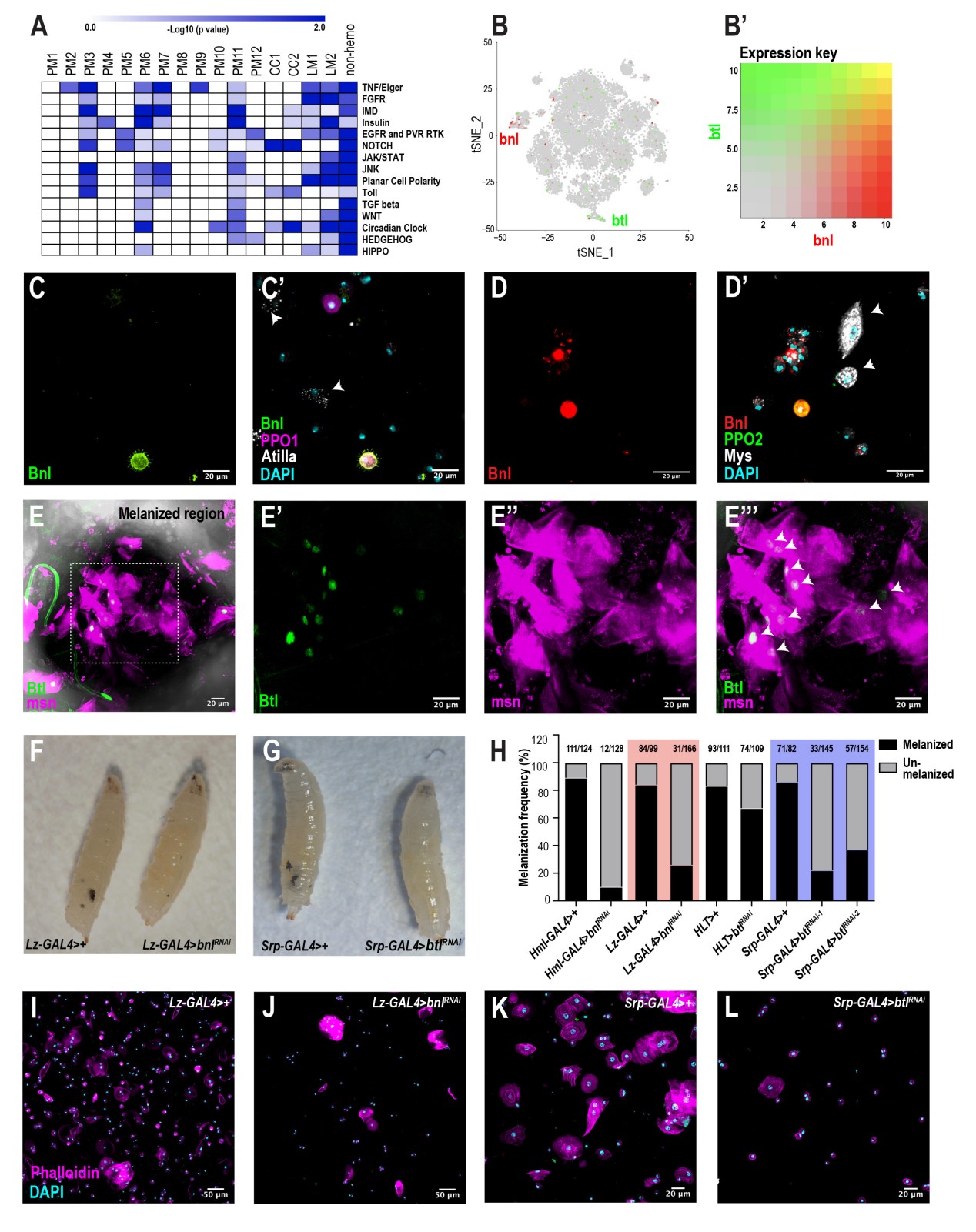

**Figure 6.** scRNA-seq uncovers a novel role for the FGF pathway in immune response. (**A**) Pathway enrichment of the top marker genes across all the clusters from *Figure 1C*. (**B**) Expression of *bnl* (red) and *btl* (green) in crystal cell and lamellocyte clusters, respectively. (**B'**) Expression heat map key. (**C**, **C'**) Validation of *bnl* expression in hemocytes of wounded *bnl-LexA; LexAOp-myr-GFP, BcF6-mCherry* larvae. Expression of GFP was detected only in crystal cells and not lamellocytes (white arrows). GFP and BcF6-mCherry represent the expression of *bnl* and *PPO1*, respectively. Scale bar = 20 μm. (**D**, *Figure 6 continued on next page*

Figure 6 continued

D') Validation of *bnl* expression in hemocytes of wasp infested *bnl-LexA; LexAOp-mCherry* larvae. Bnl (mCherry[+]), Bnl+PPO2 (red+green merged), Myospheroid (Mys, which is specific for lamellocytes) (white arrows), DAPI (cyan). Scale bar = 20 µm. (E-E''') Validation of *btl* expression in lamellocytes in vivo. The melanized region of *btl-GAL4; UAS-GFPN-lacZ, msn-mCherry* larvae was imaged using confocal microscopy. Expression of GFP was detected in LM nuclei (white arrows in E'''). GFP and msn-mCherry represent the expression of *btl* and *msn*, respectively. *msn* is a marker for lamellocytes. Scale bar = 20 µm. (F) Representative images of wasp inf. 48 hr *lz-GAL4>+* (control) and *lz-GAL4 >bnl*[RNAi] larvae. (G) Representative images of wasp inf. 48 hr *srp-GAL4>+* and *srp-GAL4 >btl*[RNAi] larvae. (H) Melanization frequencies of wasp inf. 48 hr larvae upon *bnl* and *btl* knockdown using *Hml-*, *lz-*, *HLT-*, and *srp-GAL4* drivers. (I, J) Confocal images of hemocytes from wasp inf. 48 hr larvae in *lz-GAL4>+* controls (I) compared to their *lz-GAL4 >bnl*[RNAi] (J). Scale bar = 50 µm. (K, L) Confocal images of hemocytes from wasp inf. 48 hr larvae in *srp-GAL4>+* controls (K) compared to their *srp-GAL4 >btl*[RNAi] (L). Scale bar = 20 µm.

The online version of this article includes the following figure supplement(s) for figure 6:

**Figure supplement 1.** scRNA-seq uncovers a novel role for the FGF pathway in immune response.
**Figure supplement 2.** scRNA-seq uncovers a novel role for the FGF pathway in immune response.
**Figure supplement 3.** scRNA-seq uncovers a novel role for the FGF pathway in immune response.

of *bnl* in crystal cells in uninfested conditions (*Figure 6—figure supplement 2B–C*). Moreover, because a small subset of plasmatocytes expresses *bnl*, we checked whether the recruitment of Hml[+] cells towards wasp eggs was affected upon loss of *bnl* in Hml[+] plasmatocytes. We first observed that the 10% larvae which could melanize wasp eggs in *Hml-GAL4 >bnl*[RNAi], displayed a dramatic reduction in the size of the melanotic mass (*Figure 6—figure supplement 2D–F*). In addition, the number of Hml[+] cells significantly decreased around the wasp eggs in *Hml-GAL4 >bnl*[RNAi] larvae compared to controls (*Figure 6—figure supplement 2G–I*). Interestingly, we observed a few intact crystal cells around the wasp eggs in *Hml-GAL4 >bnl*[RNAi] larvae compared to controls, which did not show any crystal cells, presumably due to the natural rupturing during infestations (*Figure 6—figure supplement 2G–H*). Whether Bnl contributes to crystal rupture remains to be seen. Together, these results suggest that Bnl, derived from crystal cells, and perhaps a subset of plasmatocytes, may play a key role in the differentiation and possibly recruitment of blood cells towards wasp eggs for effective melanization.

To address the functions of Btl in vivo, we expressed RNAi against *btl* (*btl*[RNAi]) using Hml Lineage Tracing (HLT)-GAL4 to maintain GAL4 expression in hemocytes even upon loss of *Hml* expression (*Hml-GAL4; UAS-FLP; ubi-FRT-STOP-FRT-GAL4*). After wasp infestation, *HLT-GAL4 >btl*[RNAi] larvae showed a 67% (74/109) in melanization frequency compared to that of 83% in controls (93/111) (*Figure 6H*; *Figure 6—figure supplement 3B*). This subtle decline in melanization may be due, in part, to the production of terminally differentiated lamellocytes from non-Hml[+] cells and/or possible inefficient flippase activity in Hml[+] plasmatocytes. Thus, we used the pan-hemocyte driver srp-GAL4. *Srp* is well expressed in all blood cells including lamellocytes (*Figure 1—figure supplement 1H*). As expected, srp-GAL4-mediated knockdown of *btl* using two independent RNAi lines (btl[RNAi]-1 and −2) reduced the melanization frequency to 20% and 37% (31/145 and 57/154), respectively, of wasp infested larvae compared to >85% (71/82) in control larvae (*Figure 6G,H*). Similar to *bnl*[RNAi] in crystal cells, knockdown of *btl* resulted in a significant reduction in the number of lamellocytes (*Figure 6K,L*; *Figure 6—figure supplement 3I*) with a subtle decrease in the total number of blood cells but unchanged crystal cell numbers (*Figure 6—figure supplement 3G–H*) compared to controls. Moreover, none of the cell types, including crystal cells, displayed any changes in their numbers in at least one of the two *srp-GAL4-btl*[RNAi] lines (*Figure 6—figure supplement 3D–F*). These data suggest that Btl is critical for the differentiation and possible recruitment of lamellocytes to elicit an efficient immune response upon wasp infestation. Thus, communication between Bnl[+]crystal cells and Btl[+] lamellocytes may be important in melanization of parasitoid wasp eggs.

## Discussion

Previous studies have identified three major *Drosophila* blood cell types essential for combating infections in this species (*Banerjee et al., 2019*; *Lemaitre and Hoffmann, 2007*; *Rizki, 1957*). Here, we used scRNA-seq of larval fly blood to gain deeper insights into the different cell types and their transition states in circulation during normal and inflammatory conditions. Our comprehensive scRNA-seq data provide information on subpopulations of plasmatocytes and their immune-

activated states. Importantly, our scRNA-seq could precisely distinguish mature crystal cells and lamellocytes from their respective intermediate states, which are less well understood and for which marker genes were not previously available. Thus, new marker genes identified in this study should facilitate further study of these states. Moreover, we were able to identify the gene signature of self-renewing plasmatocytes and suggest their role as extra-lymph gland oligopotent precursors (*Figure 3G*). In addition to the identification of various states of mature cell types, our study also suggests novel roles for a number of genes and pathways in blood cell biology. In particular, we identified a putative new Mtk-like AMP and proposed a role for the FGF signaling pathway in mediating key events leading to the melanization of wasp eggs (*Figure 7A*). Finally, we developed a user-friendly searchable online data mining resource that allows users to query, visualize, and compare genes within the diverse hemocyte populations across conditions (*Figure 1—figure supplement 2C*).

## Towards a fly blood cell atlas – defining cell types and states

Blood cell types are dynamic in nature and several transient intermediate states exist in a continuum during the course of their maturation in several species. Our scRNA-seq analysis provides a framework to distinguish cell types from their various states including oligopotent, transient intermediate and activated states.

### Oligopotent state

Our scRNA-seq analysis identified PM2 as the oligopotent state of plasmatocytes based on the enrichment of several cell cycle genes including *polo* and *stg*. This signature suggests that PM2 corresponds to self-renewing plasmatocytes located in the circulatory and sessile compartments of the *Drosophila* hematopoietic system where plasmatocytes are the only dividing cells identified (*Lanot et al., 2001*; *Leitão and Sucena, 2015*; *Makhijani et al., 2011*; *Rizki, 1957*). Further, previous studies suggested that lamellocytes derived from embryonic-lineage hemocytes are readily detectable in circulation prior to their release from the lymph gland (*Márkus et al., 2009*), and that terminally differentiated crystal cells can also derive from preexisting plasmatocytes in the sessile hub (*Leitão and Sucena, 2015*). Hence, we propose that PM2 corresponds to the oligopotent state that not only drives expansion of plasmatocytes, but importantly can also give rise to crystal cells and lamellocytes. Our Monocle3 analysis indicates that cell cycle genes decrease over pseudotime and there is ample evidence in support of the notion that cell cycle arrest may be required for terminal differentiation of various cell types in flies and vertebrates (*Buttitta and Edgar, 2007*; *Guo et al., 2016*; *Ruijtenberg and van den Heuvel, 2016*; *Soufi and Dalton, 2016*). Our in vivo data also indicates that cell cycle arrest can lead to the generation of terminally differentiated lamellocytes. Interestingly, recent evidence in hemocytes suggests that perturbing cell cycle by knocking down *jumu*, which is upstream of *polo*, can also lead to the generation of lamellocytes by activating Toll (*Ahmad et al., 2012*; *Hao and Jin, 2017*; *Hao et al., 2018*). In contrast, forced expression of certain oncogenes such as activated Ras and Hopscotch/JAK in hemocytes can also lead to overproduction of plasmatocytes and lamellocytes (*Arefin et al., 2017*; *Asha et al., 2003*; *Luo et al., 1995*). It is, however, speculated that the proliferation and differentiation of hemocytes in these contexts may be linked to cell cycle (*Asha et al., 2003*). Thus, it is important to address this paradoxical role of cell cycle in the maintenance of oligopotency and transdifferentiation of plasmatocytes. Studies using lineage tracing methods such as G-TRACE (*Evans et al., 2009*) or CRISPR-based in vivo cellular barcoding techniques (*Kebschull and Zador, 2018*; *Spanjaard et al., 2018*) may help further characterize the contribution of proliferating oligopotent plasmatocytes to blood cell lineages (*Figure 3G*).

### Immune-activated states

PM5 from our scRNA-seq data is enriched in several genes that encode glutathione S-transferase family of metabolic enzymes, which are known to catalyze the conjugation of reduced glutathione (GSH) to xenobiotics for their ultimate degradation (*Supplementary file 2*). It has been demonstrated that a subset of hemocytes accumulate high GSH levels in *Drosophila* (*Tirouvanziam et al., 2004*), in support of our data. Further, we classified the two AMP clusters PM6-7 (PM$^{AMP}$) as part of the immune-activated states of plasmatocytes. A recent study has demonstrated that AMPs are

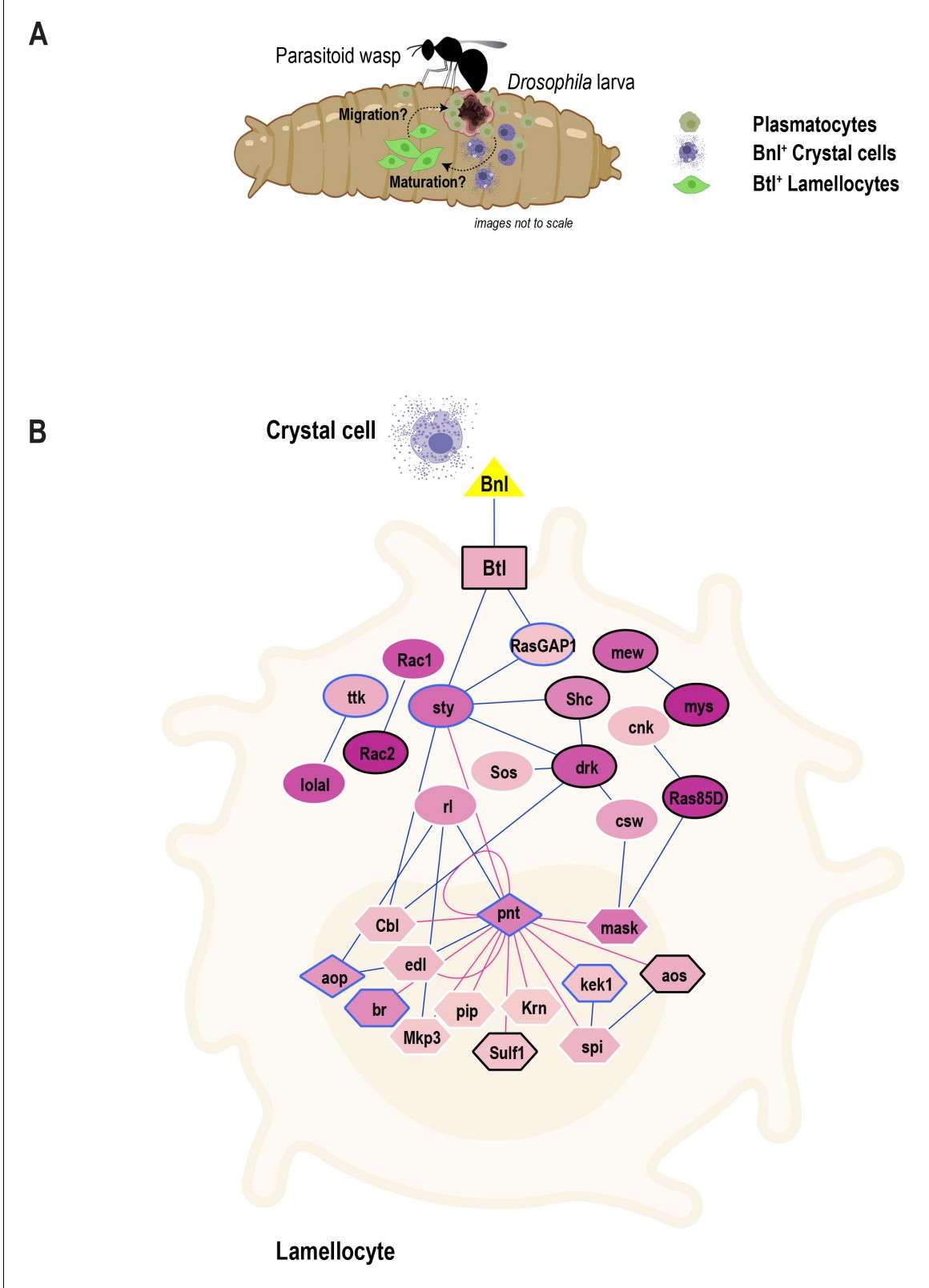

**Figure 7.** Model of the role of FGF signaling pathway in blood cell migration. (**A**) Proposed model depicting inter-hemocyte crosstalk between Bnl⁺crystal cells and Btl⁺ lamellocytes. Based on our in vivo data, we propose that crystal cells expressing *Bnl* are important for the differentiation or maturation and possible migration or recruitment of lamellocytes towards parasitoid wasp eggs. (**B**) FGF signaling pathway map depicting the enrichment of genes that encode core components of the FGF signaling pathway. Triangle, rectangle, circles, diamonds, polygons represent ligand,

*Figure 7 continued on next page*

*Figure 7 continued*

receptor, signaling proteins, transcription factors, and their downstream target genes, respectively. Color gradient within nodes represents the number of cells a particular gene is enriched. Blue line (edge color) represents protein-protein interactions (from PPI network) and red line represents transcription factor-target gene networks. Genes that are more enriched in lamellocytes are in black border and those that are in blue border represent less enriched genes. Note that some genes with white border may be enriched in other clusters and are not marker genes of LM2.

highly specific and act in synergy against various pathogens (*Hanson et al., 2019*). Our scRNA-seq analysis reveals the remarkable difference in the expression of a set of AMPs in the two clusters. Future studies with PM$^{AMP}$-specific perturbation of various AMPs identified within plasmatocytes should clarify their contribution in killing specific pathogens. Moreover, the role of *Mtkl* against pathogens needs further characterization. Our pseudotime analysis showed that PM$^{AMP}$ ends in the same lineage as lamellocytes suggesting a common mode of activation for these cell types and states. Interestingly, induction in hemocytes of Toll, which is upstream of *Drs*, can lead to the production of lamellocytes (*Hao et al., 2018*; *Schmid et al., 2014*; *Zettervall et al., 2004*), suggesting that LM$^{int}$ cells may act as the common branch point between immune-activated states and lamellocytes.

## Transient intermediate states

In addition to the oligopotent and immune-activated states, plasmatocytes showed several subpopulations, which most likely are transient intermediate states. Although it remains to be seen whether they exist throughout the larval development, it is possible that these transient states exist along the continuum of cell maturation process. On the other hand, the transcriptomic composition of CC1 and LM1 clusters suggested the presence of intermediates for crystal cells and lamellocytes, respectively. Further analysis by Monocle3, which placed these clusters prior to their terminally differentiated cell types, confirmed our hypothesis that CC1 and LM1 correspond to CC$^{int}$ and LM$^{int}$ states, respectively. In the context of the CC lineage within the lymph gland, ultrastructural studies have revealed the presence of immature crystal cells, called procrystal cells, alongside mature crystal cells (*Shrestha and Gateff, 1982*). We furthered this observation by demonstrating in vivo that crystal cells exist in a continuum (PPO1$^{low}$ to PPO1$^{high}$), validating our Monocle3 and scRNA-seq data. Moreover, clear gene signatures between the CC$^{int}$ and LM$^{int}$ states and their mature counterparts revealed that these intermediates most likely emerge from preexisting Hml$^+$ plasmatocytes. With regards to the LM lineage, several groups have speculated that intermediates, called podocytes, or also lamelloblasts, may exist based on cell morphology and size (*Anderl et al., 2016*; *Brantley et al., 2018*; *Rizki, 1957*; *Rizki, 1962*). Our scRNA-seq and Monocle3-based data clearly demarcate mature lamellocytes from LM$^{int}$ at the transcriptomic level. In addition, our sub-clustering analysis revealed that LM$^{int}$ possessed a PM signature demonstrating that these intermediates are presumably derived from PM2.

## A novel role for the FGF signaling pathway in hemocyte crosstalk

In addition to the known hemocyte – tissue crosstalk (*Shia et al., 2009*), *Drosophila* hemocytes must act in a coordinated fashion to combat harmful pathogens and foreign entities such as wasp eggs (*Banerjee et al., 2019*; *Lemaitre and Hoffmann, 2007*). However, the signaling pathways that mediate the interactions among hemocytes and wound sites or wasp eggs have been unclear. Our scRNA-seq uncovered a novel role for the FGF signaling pathway in controlling hemocyte differentiation and subsequent effects on the melanization of wasp eggs. The FGF ligand *bnl* and its receptor *btl* were among the genes identified in rare subsets of crystal cells and lamellocytes, respectively, highlighting the power of scRNA-seq in capturing and detecting these small populations of cells. Based on our in vivo data, we propose that Bnl$^+$crystal cells interact with Btl$^+$ lamellocytes to coordinate lamellocyte differentiation and possible migration towards parasitoid wasp eggs (*Figure 7A*). Furthermore, because lamellocytes are also enriched in additional core components of the FGF signaling pathway (*Figure 7B*), future studies involving a comprehensive analysis of this pathway will advance our understanding of blood cell communication, differentiation, and migration in the context of immune response.

In summary, our scRNA-seq data provides a resource for a comprehensive systems-level understanding of *Drosophila* hemocytes across various inflammatory conditions.

# Materials and methods

## Key resources table

| Reagent type (species) or resource | Designation | Source or reference | Identifiers | Additional information |
|---|---|---|---|---|
| Gene (*D. melanogaster*) | CG43236 | NA | FLYB: FBgn 0262881 | NA |
| Gene (*D. melanogaster*) | E(spl)m3-HLH | NA | FLYB: FBgn 0002609 | NA |
| Gene (*D. melanogaster*) | Drip | NA | FLYB: FBgn 0015872 | NA |
| Gene (*D. melanogaster*) | bnl | NA | FLYB: FBgn 0014135 | NA |
| Gene (*D. melanogaster*) | btl | NA | FLYB: FBgn 0285896 | NA |
| Strain, strain background (*D. melanogaster*) | Oregon R | Bloomington *Drosophila* Stock Center | BDSC: 5 | NA |
| Strain, strain background (*L. boulardi*) | Leptopilina boulardi | Bloomington *Drosophila* Stock Center | PMID:17967061 | Strain G486 |
| Genetic reagent (*D. melanogaster*) | HmlΔ-GAL4; UAS-2XEGFP | Bloomington *Drosophila* Stock Center | BDSC: 30140; FLYB: FBst 0030140 | FLYB genotype: w1118; P{Hml-GAL4.Δ}2, P{UAS-2xEGFP}AH2 |
| Genetic reagent (*D. melanogaster*) | Hml-GAL4-Lineage Trace (HLT)-GAL4 | Dr. Utpal Banerjee | PMID:22134547 | *Hml-Gal4 UAS-FLP, ubi-FRT-STOP-FRT-Gal4* |
| Genetic reagent (*D. melanogaster*) | lz-GAL4; UAS-GFP | Bloomington *Drosophila* Stock Center | BDSC: 6314; FLYB: FBst 0006314 | FLYB genotype: y1 w* P{UAS-mCD8::GFP.L}Ptp4ELL4 P{GawB}lzgal4 |
| Genetic reagent (*D. melanogaster*) | E(spl)m3-HLH-GAL4 | Bloomington *Drosophila* Stock Center | BDSC: 46517; FLYB: FBst 0046517 | FLYB genotype: w1118; P{GMR10E12-GAL4}attP2 |
| Genetic reagent (*D. melanogaster*) | Drip-GAL4 | Bloomington *Drosophila* Stock Center | BDSC: 66782; FLYB: FBst 0066782 | FLYB genotype: y1 w*; Mi{Trojan-GAL4.0} DripMI00887-TG4.0/SM6a |
| Genetic reagent (*D. melanogaster*) | btl-GAL4 | Perrimon Lab stock | NA | Genotype: yw; UAS-GFPN-lacZ(2-1)/CyO; btl-Gal4(3-1)/TM3 Sb Ser |
| Genetic reagent (*D. melanogaster*) | bnl-LexA | Dr. Sougata Roy | PMID:28502613 | *bnl-LexA/TM6* |
| Genetic reagent (*D. melanogaster*) | Ubi-GAL4 | Dr. Utpal Banerjee | PMID:24267893 | *Ubiquitin-GAL4* |
| Genetic reagent (*D. melanogaster*) | BcF6-mCherry | Dr. Robert Schulz | PMID:27913635 | BcF6-mCherry (III) |
| Genetic reagent (*D. melanogaster*) | msn-mCherry | Dr. Robert Schulz | PMID:27913635 | MSNF9mo-mCherry (III) |
| Genetic reagent (*D. melanogaster*) | srp-GAL4 | Dr. Lucas Waltzer | PMID:14657024 | NA |
| Genetic reagent (*D. melanogaster*) | LexAOp-myr-GFP | Bloomington *Drosophila* Stock Center | BDSC:32210; FLYB: FBst 0032210 | FLYB genotype: P{13XLexAop2-IVS-myr::GFP}attP40 |

*Continued on next page*

*Continued*

| Reagent type (species) or resource | Designation | Source or reference | Identifiers | Additional information |
|---|---|---|---|---|
| Genetic reagent (*D. melanogaster*) | LexAOp-mCherry | Bloomington *Drosophila* Stock Center | BDSC:52271; FLYB: FBst 0052271 | FLYB genotype: y1 w*; wgSp-1/CyO, P{Wee-P.ph0}BaccWee-P20; P{13XLexAop2-6XmCherry-HA}attP2 |
| Genetic reagent (*D. melanogaster*) | UAS-mCD8-GFP | Bloomington *Drosophila* Stock Center | BDSC: 5137; FLYB: FBst 0005137 | FLYB genotype: y1 w*; P{UAS-mCD8::GFP.L} LL5, P{UAS-mCD8::GFP.L}2 |
| Genetic reagent (*D. melanogaster*) | UAS-polo^RNAi | Bloomington *Drosophila* Stock Center | BDSC: 33042; FLYB: FBst 0033042 | FLYB genotype: y1 sc* v1 sev21; P{TRiP.HMS00530}attP2 |
| Genetic reagent (*D. melanogaster*) | UAS-btl^RNAi-1 | Bloomington *Drosophila* Stock Center | BDSC: 43544; FLYB: FBst 0043544 | FLYB genotype: y1 sc* v1 sev21; P{TRiP.HMS02656}attP40 |
| Genetic reagent (*D. melanogaster*) | UAS-btl^RNAi-2 | Bloomington *Drosophila* Stock Center | BDSC: 60013; FLYB: FBst 0060013 | FLYB genotype: y1 v1; P{TRiP.HMS05005} attP40 |
| Genetic reagent (*D. melanogaster*) | UAS-empty | Dr. Hugo Bellen | PMID:27640307 | UAS-empty (III) |
| Transfected construct (*D. melanogaster*) | NA | NA | NA | NA |
| Biological sample (*D. melanogaster*) | larval hemolymph (blood) | NA | NA | Hemocytes from hemolymph of third instar (96 and 120 hr AEL) larvae |
| Antibody | L1abc (mouse monoclonal) | Prof. Istvan Andó | PMID:18297797 | 1:100 dilution |
| Antibody | anti-PPO2 (mouse monoclonal) | Prof. Istvan Andó | PMID:18297797 | 1:1000 dilution |
| Antibody | anti-Hindsight (mouse monoclonal) | Developmental Studies Hybridoma Bank | Cat# 1G9 | 1:10 dilution |
| Antibody | anti-Mys (mouse monoclonal) | Developmental Studies Hybridoma Bank | Cat# CF-6G11 | 1:10 dilution |
| Recombinant DNA reagent | NA | NA | NA | NA |
| Sequence-based reagent | CecC (primer) | FlyPrimer Bank | PD41779 | For: GCATTGGACAATCGGAAGCC Rev: TTGCGCAATTCCCAGTCCTT |
| Sequence-based reagent | Drs (primer) | FlyPrimer Bank | PD40133 | For: CTGGGACAACGAGACCTGTC Rev: ATCCTTCGCACCAGCACTTC |
| Sequence-based reagent | Mtk (primer) | FlyPrimer Bank | PD41985 | For: GCTACATCAGTGCTGGCAGA Rev: TTAGGATTGAAGGGCGACGG |
| Sequence-based reagent | CG43236 (primer) | FlyPrimer Bank | PD41670 | For: GCAAGAGTTT GGATGCCACC Rev: GCCTCATATCG AAAGGATTGCG |
| Sequence-based reagent | stg (primer) | FlyPrimer Bank | PB60117 | For: GAAAACAACTG CAGCATGGAT Rev: CGACAGCT CCTCCTGGTC |

*Continued on next page*

Continued

| Reagent type (species) or resource | Designation | Source or reference | Identifiers | Additional information |
|---|---|---|---|---|
| Sequence-based reagent | polo (primer) | FlyPrimer Bank | PP7029 | For: CCCGAGGATA AGAGCACGGA Rev: GTCGTCGGTTTCCACATCG |
| Sequence-based reagent | MMP1 (primer) | FlyPrimer Bank | PP18419 | For: CCAGTTCGGCTATCTACCCG Rev: CTCGATGGCACTCACCCAG |
| Sequence-based reagent | Ance (primer) | FlyPrimer Bank | PP22471 | For: GTGATACCACCA AGTTCCAATGG Rev: GGCATAGTCGT CTTCAGGTAGAG |
| Sequence-based reagent | GstE6 (primer) | FlyPrimerBank | PP10905 | For: TACGGTTTGGACCCCAGTC Rev: ATATTCCGGTGAAAGTTGGGC |
| Sequence-based reagent | Arc1 (primer) | FlyPrimerBank | PP10071 | For: ATGGCCCAGCTTACACAGATG Rev: GGAGAAGTTGCCTTTGCCTC |
| Sequence-based reagent | Prx2540-1 (primer) | FlyPrimerBank | PD40349 | For: ATGATCCTGCCCACTGTCAC Rev: CAGTGGTGCGGACGTAGTTT |
| Sequence-based reagent | Ubx (primer) | FlyPrimerBank | PP12922 | For: ATGAACTCGTA CTTTGAACAGGC Rev: CCAGCGAGA GAGGGAATCC |
| Sequence-based reagent | Cpx (primer) | FlyPrimerBank | PD40622 | For: CGCGAGAAGA TGAGGCAAGA Rev: CATCAGGGGA TTGGGCTCTT |
| Sequence-based reagent | mthl7 (primer) | FlyPrimerBank | PP15001 | For: AGTTTGGGGA CGGTTCGATTA Rev: TGAGACCATCA TCGCATTTTCC |
| Sequence-based reagent | Cys (primer) | FlyPrimerBank | PP22082 | For: GGATGCCAC TCTCGCACAG Rev: GGTGTTAAGA CTTCCAGCTACG |
| Sequence-based reagent | bnl (primer) | NA | NA | For: AACCCAAATCCAATCCCAAT Rev: GATGCTGTTGCTGTTGCTGT |
| Sequence-based reagent | btl (primer) | NA | NA | For: GAGTCGATCCCTGAAGTTGC Rev: GCAGTTGCCCCACTGTTAAT |
| Sequence-based reagent | RpL32/rp49 (primer) | FlyPrimerBank | PD41810 | For: AGCATACAGGCCCAAGATCG Rev: TGTTGTCGATACCCTTGGGC |
| Peptide, recombinant protein | NA | NA | NA | NA |
| Commercial assay or kit | Chromium Single Cell 3' Library and Gel Bead Kit v2 | 10x Genomics | PN-120267 | NA |
| Commercial assay or kit | Chromium i7 Multiplex Kit | 10x Genomics | PN-120262 | NA |
| Commercial assay or kit | Chromium Single Cell A Chip Kit | 10x Genomics | PN-1000009 | NA |
| Chemical compound, drug | NA | NA | NA | NA |
| Software, algorithm | Seurat | *Stuart et al., 2019* | PMID:31178118 | NA |
| Software, algorithm | Harmony | *Korsunsky et al., 2019* | PMID:31740819 | NA |

*Continued*

| Reagent type (species) or resource | Designation | Source or reference | Identifiers | Additional information |
|---|---|---|---|---|
| Software, algorithm | Monocle 3 | *Cao et al., 2019* | PMID:30787437 | NA |
| Software, algorithm | Jalview | *Waterhouse et al., 2009* | PMID:19151095 | NA |
| oftware, algorithm | Biorender | https://biorender.com/ | NA | Biorender was utilized to make the schematic diagrams used in this study. |
| Other | DAPI (nuclear stain) | Vector Laboratories | Cat# H-1200 | Ready to use |
| Other | Phalloidin | ThermoFischer | Cat# A34055 | 1:100 dilution |
| Other | Optiprep | AxisShield | AXS-1114542 | Working concentration: 1.09 g/ml |
| Other | SyBr Green | Bio-Rad iQ SYBR Green Supermix | Cat# 1708880 | Working concentration: 1X |

## Fly stocks and reagents

*Drosophila melanogaster* larvae of the genetic backgrounds *w;Hml-GAL4Δ, UAS-2X EGFP* (*Hml > EGFP*) or *Oregon R* (*OreR*) were used for the preparation of single hemocytes. Third instar *Hml > EGFP* larvae and second instar *OreR* larvae were used for wounding and wasp infestations, respectively. To visualize the crystal cell hubs, yw; *lz-GAL4; UAS-mCD8::GFP* (*lz >GFP*) (BL# 6314) flies were crossed to *w;;BcF6-mCherry* flies and the resultant female larvae positive for both reporters were used for confocal microscopy. The following stocks were obtained from the Bloomington *Drosophila* Stock Center (BDSC), GAL4 lines: Drip-GAL4 (BL# 66782), E(spl)m3-HLH-GAL4 (BL# 46517); RNAi lines: bnl-RNAi (BL# 34572), btl-RNAi (BL# 43544; 60013), polo-RNAi (BL# 33042); and Reporter line: LexAOp-myr-GFP (BL# 32210). The srp-GAL4 was obtained from a previous study (*Waltzer et al., 2003*). The bnl-LexA line was a kind gift from Dr. Sougata Roy (*Du et al., 2017*). The yw;UAS-GFPN-lacZ;btl-GAL4 (which expresses a nuclear GFP/lacZ fusion protein under the control of btl-GAL4) and w;Hml-GAL4Δ, UAS-2X EGFP lines are Perrimon Lab stocks. BcF6-mCherry and msn-mCherry fly stocks were obtained from Dr. Robert Schulz (*Tokusumi et al., 2017*).

The *lz-GAL4; UAS-mCD8::GFP, BcF6-mCherry* line was obtained by crossing *lz-GAL4; UAS-mCD8::GFP* with *BcF6-mCherry* flies. The *bnl-LexA; LexAOp-myr-GFP, BcF6-mCherry* line (Bnl >GFP; BcF6-mCherry) was obtained by crossing *bnl-LexA; LexAOp-myr-GFP* and *BcF6-mCherry* flies. The *btl-GAL4, UAS-GFPN-lacZ, msn-mCherry* line was obtained by crossing *btl-GAL4, UAS-GFPN-lacZ* to *msn-mCherry* reporter line.

All flies and larvae were maintained on standard fly food at 25°C.

## Antibodies

Antibodies against Atilla [anti-mouse L1abc (1:100 dilution)] and PPO2 (1:1000 dilution) were generous gifts from István Ando (*Kurucz et al., 2007*). Phalloidin (ThermoFischer A34055) was used at a concentration of 1:100. Hindsight (Hnt; 1:10 dilution) and Myospheroid (Mys; 1:10 dilution) antibodies were obtained from DSHB.

## Wounding and wasp infestation

### Wounding

Precisely timed 24 hr (hours) after egg lay (AEL) larvae of the *hml >EGFP* genotype were collected and grown on normal fly food until they reached 96 hr AEL for wounding procedure. At 96 hr AEL, larvae were either left unwounded or wounded with a clean tungsten needle (Fine Science Tools, cat# 10130–05). Ten larvae at a time were wounded at their posterior dorsal side and returned back

to fly food with a total of at least 80 wounded larvae per vial. 24 hr later, the unwounded control and wounded larvae were retrieved from fly food and washed in distilled water twice.

### Wasp infestation

*OreR* larvae were infested at 72 hr AEL with the wasps of the species *Leptopilina boulardi*, strain G486. Wasps were removed after 12 hr of co-culture and egg deposition was confirmed by direct observation of wasp eggs in the hemolymph during dissection. 100 larvae were dissected at 96 and 120 hr AEL, corresponding to 24 and 48 hr post infestation (wasp inf. 24 hr and 48 hr), respectively, in Schneider's medium (Gibco, cat# 21720024).

## Preparation of single hemocytes in suspension

To get most of the sessile hemocytes into circulation, washed larvae were transferred to 2 ml Eppendorf tubes containing ~0.5 ml of glass beads (Sigma #9268, size: 425–600 µm) in PBS and larvae were vortexed for 2 min as previously described with minor modifications (*Petraki et al., 2015*). One set each of unwounded and wounded larvae were vortexed in separate tubes at a time. After the brief vortex, larvae were retrieved, washed and transferred to 200 µl of ice-cold PBS in each well of a clean 9-well glass dish per condition. ~100 larvae were bled by gently nicking open the posterior side of each larva using a pair of clean tweezers. Larvae were allowed to bleed for at least a minute and the hemolymph in PBS was filtered through 100 µm cell strainer and the filtered hemolymph was overlaid onto 2 ml of 1.09 g/ml Optiprep gradient solution (Axis-Shield cat# AXS-1114542) and spun at 2000 rpm for 30 min at 4°C to eliminate dead cells and debris. After centrifugation,~150 µl of the hemolymph was transferred to clean low bind Eppendorf tubes and counted using a hemocytometer. High quality single hemocytes were subjected to encapsulation either by inDrops or 10X Genomics v2 platform.

Hemocytes from wasp infested larvae were isolated with some modifications, where the optiprep step was avoided to obtain higher number of cells. Briefly, after vortexing, the larvae were bled in ice cold Schneider's medium, filtered through 100 µm cell strainer, and transferred to a clean Eppendorf tube. Next, the cells were spun at 4°C for 5 min at 6000 rpm. The supernatant was discarded, and the cells were re-suspended in ice cold PBS to achieve a concentration of 300 cells/µl and subjected to Drop-seq based encapsulation.

## Single hemocyte encapsulation and sequencing

Single hemocytes from unwounded control (n = 4) and wounded larvae (n = 4), respectively, were encapsulated either by inDrops or 10X genomics v2 platforms, with n = 2 per platform.

For inDrops, hemocytes were encapsulated at the Single Cell Core facility of the ICCB-Longwood Screening Facility at Harvard Medical School (https://singlecellcore.hms.harvard.edu/) using the inDrops v3 library format (*Klein et al., 2015*). Reverse transcription and library preparation were performed at the same facility. The libraries were made following a previously described protocol (*Klein et al., 2015*; *Zilionis et al., 2017*), with the following modifications in the primer sequences:

RT primers on hydrogel beads: 5' – CGATTGATCAACGTAATACGACTCACTATAGGGTGTCGGG TGCAG [bc1,8nt] GTCTCGTGGGCTCGGAGATGTGTATAAGAGACAG [bc2,8nt] NNNNNNTTTTTTTTTTTTTTTTTTTTTV – 3'. R1-N6 primer sequence [step 151 in the library prep protocol (*Zilionis et al., 2017*): 5' – TCGTCGGCAGCGTCAGATGTGTATAAGAGACAGNNNNNN – 3'. PCR primer sequences (steps 157 and 160 in the library prep protocol in *Zilionis et al., 2017*: 5' – AATGATACGGCGACCACCGAGATCTACACXXXXXXXXXTCGTCGGCAGCGTC – 3', where XXXXXX is an index sequence for multiplexing libraries. 5' – CAAGCAGAAGACGGCATACGAGATGGGTG TCGGGTGCAG – 3' With these modifications in the primer sequences, custom sequencing primers are no longer required.

The fragment size of each library was analyzed using a Bioanalyzer high sensitivity chip. Libraries were diluted to 1.5 nM and then quantified by qPCR using primers against the p5-p7 sequence. inDrops libraries were sequenced on an Illumina Nextseq 500 with following parameters: (1) Read 1: 61 cycles, (2) i7 index: 8 cycles index 1, (3) i5 index: 8 cycles (i5), and (4) Read 2: 14 cycles. Binary base call (BCL) files were converted into FASTQ format with bcl2fastq, using the following flags that are required for inDrops v3: (1) `–use-bases-mask y*,y*,y*,y*`; (2) `–mask- short-adapter-reads` 0; (3) `–minimum-trimmed-read-length` 0.

With regards to 10X genomics, cells were encapsulated according to the manufacturer's protocol. cDNA libraries generated by both platforms were sequenced after pooling four different (indexed) samples per one 400M plus NextSeq500 cartridge with following parameters: (1) Read 1: 26 cycles, (2) i7 index: 8 cycles, (3) i5 index: 0 cycles, and (4) Read 2: 57 cycles.

Drop-seq was used to encapsulate the wasp inf. 24 hr samples (n = 3). The protocol for Drop-seq based encapsulation was followed as previously described (*Macosko et al., 2015*).

## Data processing

### Count matrix generation inDrops:

The software version used to generate counts from the FASTQ files were managed with bcbio-nextgen 1.0.6a0-d2b5b522 (https://github.com/bcbio/bcbio-nextgen) using bioconda (*Grüning et al., 2018*) (https://bioconda.github.io/). First, cellular barcodes and UMIs were identified for all reads. Second, reads with one or more mismatches of a known barcode were discarded. Third, remaining reads were quasi-aligned to the FlyBase FB2018_02 Dmel Release 6.21 reference transcriptome using RapMap 0.5.0 (*Srivastava et al., 2016*) (https://github.com/COMBINE-lab/RapMap). EGFP and GAL4 sequences were included in the transcriptome as spike-in genes (https://github.com/hbc/A-single-cell-survey-of-Drosophila-blood; *Steinbaugh et al., 2020*; copy archived at https://github.com/elifesciences-publications/A-single-cell-survey-of-Drosophila-blood) Reads per cell were counted using the umis 0.6.0 package for estimating UMI counts in transcript tag counting data (*Svensson et al., 2017*) (https://github.com/vals/umis), discarding duplicated UMIs, weighting multi-mapped reads by the number of transcripts they aligned to, and collapsing counts to genes by adding all counts for each transcript of a gene. Finally, cellular barcodes with fewer than 1000 reads assigned were discarded from the analysis (see 'minimum_barcode_depth' in bcbio documentation for details).

### 10x genomics

BCL files were analyzed with the Cell Ranger pipeline (v2.1.1). The demultiplexed FASTQ data were aligned to FlyBase FB2018_02 Dmel Release 6.21 reference to generate the single cell count matrix.

### Drop-seq

Paired-end reads were processed and mapped to the reference genome BDGP 6.02 (Ensembl September 2014) following the Drop-seq Core Computational Protocol version 1.2 (January 2016) and corresponding Drop-seq tools version 1.13 (https://github.com/broadinstitute/Drop-seq) (December 2017) provided by McCarroll Lab (http://mccarrolllab.org/dropseq/). The Picard suite (https://github.com/broadinstitute/picard) was used to generate the unaligned bam files which were processed using the *Drop-seq_alignment.sh* script. The steps include detection of barcode and UMI sequences, filtration and trimming of low-quality bases and adaptors or poly-A tails, and alignment of reads using *STAR* (2.5.3a).

### Quality control (QC) analysis and filtering inDrops:

Gene-level counts were imported to R using the bcbioSingleCell 0.1.15 package (https://github.com/hbc/bcbioSingleCell). This package extends the Bioconductor SingleCellExperiment container class, which is optimized for scRNA-seq (*Huber et al., 2015*). *SingleCellExperiment: S4 Classes for Single Cell Data*. R package version 1.6.0.] (https://bioconductor.org/packages/SingleCellExperiment). QC analysis was performed using this package, and the 'filterCells()' function was used to filter out low quality cells by keeping cellular barcodes with the following metrics: (1)>=100 UMIs per cell; (2)>=100 genes per cell; (3)>=0.85 novelty score, calculated as log10(genes detected)/log10(UMI counts per cell). Additionally, genes with very low expression across the data set were filtered by applying a cutoff of >= 10 cells per gene. One sample, blood3_TCGCATAA, was filtered at a higher threshold of 650 genes per cell, which was required to subset the input cellular barcodes into the expected biological range based on the inDrops encapsulation step.

### 10x genomics

QC was performed by keeping cells with the following metrics: (1)>=500 UMIs per cell; (2)>=200 genes per cell; (3)<=30 percentage of mitochondria genes.

### DropSeq

Cumulative distribution of reads from the aligned bam files were obtained using *BAMTagHistogram* from the Drop-seq tools package. The number of cells were inferred from the sharp decrease in the slope. The inferred cell number was determined as a minimal threshold number of aligned reads per cell for cell selection.

In our QC pipeline, we did not regress out cell cycle genes during clustering for two reasons: 1. Cell cycle genes did not contribute to the variation for downstream clustering, and 2. We expected cycling plasmatocytes to be an important aspect in our analysis. Hence, cell cycle parameters were retained throughout clustering process.

### Data integration

For combined analysis of all samples, the quality filtered datasets were merged using the common genes into a single Seurat (version 3.1) object (*Stuart et al., 2019*) and integrated using Harmony (*Korsunsky et al., 2019*). The gene symbols from inDrops, 10X, and Drop-seq were converted to the same version by using FlyBase online ID Converter tool (FB2019_03) (*Thurmond et al., 2019*). Genes expressed in at least two out of three conditions were retained when combining datasets from different technologies, to minimize loss of genes, as Harmony uses common genes across all conditions. We ran PCA using the expression matrix of the top 2000 most variable genes. The total number of principal components (PCs) to compute and store were 20. Theta values were set c(2, 10) for condition and technology. A resolution of 0.4 was chosen as clustering parameter. The t-SNE was then performed using default parameters to visualize data in the two-dimensional space.

### Merging of clusters

At 0.4 resolution, 20 clusters were obtained and three clusters (1, 14, and 19) shared similar gene expression with that of cluster 0. Hence, we merged clusters 1, 14, and 19 into 0 (see *Figure 1—figure supplement 1F*).

### Sub-clustering

Crystal cell sub-clustering and lamellocyte sub-clustering follow the same procedure but with different parameters. For Crystal cell sub-clustering we used theta = c(2, 5) for condition and technology and clustering resolution of 0.1. For lamellocyte sub-clustering we used theta = c(3, 8) for condition and technology and clustering resolution of 0.1.

### Gene Expression Visualization by dot plots

Dot plots were generated using Seurat DotPlot function. Heatmaps were generated using Seurat DoHeatmap function and split by condition.

## Pseudotemporal ordering of cells using Monocle3

Cells from the unwounded and wounded data sets (10X platform; n = 2 per condition) were analyzed by Monocle3 (https://github.com/cole-trapnell-lab/monocle3; *Cao et al., 2019*). The input data set was pre-processed by using num_dim equal 65. Then the data was applied by align_cds function to remove the batch effect. The cell trajectory was calculated by using learn_graph function and ncenter equal 1000. Three lineages were selected by using choose_graph_segments function. The gene expression along pseudotime data were extracted from the result of plot_genes_in_pseudotime function. Then the data was used to plot genes along pseudotime in three lineages using ggplot2 v3.2.1 R package and the heatmap was generated using pheatmap v1.0.12 R package. The Ridgeline plot was generated using ggridges v0.5.1 R package.

### Assignment of the start point

To be unbiased, we calculated the average expression of three cell cycle associated genes enriched in PM2: *polo*, *stg*, and *scra*. We then assigned the start point, which coincidentally also overlapped with the high expression of *Hml* around the same start point.

## Data files and analysis code

The original FASTQ files, UMI-disambiguated counts in MatrixMarket Exchange format (MTX files) [see https://math.nist.gov/MatrixMarket/info.html for details], and inDrops v3 sample barcodes used are available on the NCBI Gene Expression Omnibus (GEO) with the accession number GSE146596. The code used to perform clustering and marker analysis is available on GitHub at https://github.com/hbc/A-single-cell-survey-of-Drosophila-blood.

## Gene set enrichment analysis

We performed gene set enrichment analysis on marker genes with positive fold change for each cluster using a program written in-house. Gene sets for major functional groups were collected from the GLAD database (*Hu et al., 2015*), and gene sets for metabolic pathways were from the KEGG database (*Kanehisa et al., 2017*) and Reactome database (*McKay and Weiser, 2015*). P-value enrichment was calculated based on the hypergeometric distribution. The strength of enrichment was calculated as negative of log10(p-value), which is used to plot the heatmap.

With regard to the FGFR pathway analysis in *Figure 7B*, all the interactions were based on protein-protein interaction data retrieved from molecular interaction search tool (MIST) release 5 (*Hu et al., 2018*). The expression pattern in CC2 and LM2 was analyzed. The receptor-ligand pair (Btl-Bnl) where the receptor expressing in at least 5 cells of LM2 cluster while ligand expressing in at least 5 cells of CC2 cluster was selected. The core components of the FGFR signaling pathway were retrieved from GLAD (*Hu et al., 2015*) while the annotation of transcription factor (TF) - target genes was obtained from manual curation of a related publication (database not published). The protein-protein interaction data among the pathway components were obtained from MIST and the network was built using Cytoscape v3.2.0 (*Franz et al., 2016*). Node color reflects the number of cells in LM2 cluster (the gene nodes with <5 cells in LM2 were removed from network). Node shapes reflect the ligands, receptors, TFs or their target genes.

## Immunostaining and confocal microscopy

### Whole larval imaging

Third instar *lz-GAL4 >GFP; BcF6-mCherry* larvae were heat killed in glycerol on a glass slide and directly mounted onto coverslip-bottom imaging dishes (ibidi, cat# 81158). The posterior hemocyte hubs were imaged with Z stacks of 2 µm distance each encompassing the entire area of the hub in a lateral direction using a Zeiss LSM 710 confocal microscope. Finally, all the stacks per larva were merged by summing the intensities on Fiji software for subsequent intensity measurements of GFP and mCherry.

Wasp infested *w;UAS-GFPN-lacZ;btl-GAL4* larvae were imaged using Nikon C2 Si-plus confocal microscope.

### Staining hemocytes from unwounded or wounded larvae

~13 larvae per condition or genotype were vortexed and bled into 300 µl of Schneider's insect media and then the cells were transferred to one well (per condition) of the Lab-Tek II chambered coverglass (VWR, cat# 62407-056). Hemocytes were allowed to settle down for 30 min, then were fixed in 4% paraformaldehyde (PFA) for 20 min. Next, the cells were washed three times with 1X PBS and permeabilized with 0.1% PBST (PBS with 0.1% Triton-X) for 10 min. Subsequently, the cells were blocked with 5% BSA in PBST (blocking buffer) for 20 min and subsequently incubated with primary antibodies L1abc (1:100 dilution) overnight at 4°C. The next day, cells were washed three times in PBST and incubated with corresponding secondary antibodies (1:500 dilution) for 1 hr at room temperature. Finally, the cells were washed (3X) and mounting media with DAPI was directly added onto the cells in the wells and imaged using Nikon Spinning disk microscope.

### Staining hemocytes from wasp infested larvae

Larvae were vortexed with glass beads for one minute before bleeding to detach sessile hemocytes. Next, the larvae were bled on a slide glass (Immuno-Cell Int. cat# 2018A29) and hemocytes allowed to settle onto the slide at 4°C for 40 min. Hemocytes were washed 3 times in 0.4% Triton X-100 in 1x PBS for 10 min and blocked in 1% BSA/0.4% TritonX in 1xPBS for 30 min. Primary antibody was added, and samples incubated overnight at 4°C. Hemocytes were washed 3 times in 0.4% Triton X in

1xPBS and then incubated with a secondary antibody with 1% BSA/0.4% Triton X in 1xPBS for 3 hr at room temperature. After washing 3 times with 0.4% Triton X in 1xPBS, samples were kept in Vectashield (Vector Laboratory) with DAPI and imaged by a Nikon C2 Si-plus confocal microscope.

## Cell counting

Hemocytes were mounted and imaged by Nikon C2 Si-plus or Zeiss Axiocam 503. Captured image of hemocytes were quantified and analyzed by ImageJ (plugin: 3D object counter) or Imaris (Bitplane). Hemocytes bled from individual whole larvae were counted for this study.

## Quantitative real time polymerase chain reaction (qRT-PCR)

Hemolymph with hemocytes was derived 24 hr after wounding along with their unwounded control *Hml >EGFP* larvae. 50 ~ 80 larvae per biological replicate were bled in 80 µl of Schneider's insect media and the resulting hemolymph (~100 µl) was transferred to RNase free Eppendorf tubes and frozen on dry ice. In parallel, whole larvae (four larvae per biological replicate) were directly transferred to the Eppendorf tubes and frozen on dry ice. For RNA isolation from hemocytes, 1 ml of TRIzol (ThermoFischer cat# 15596–026) was added to each sample and incubated for 5 min before adding 0.2 ml of chloroform for subsequent phase separation. The tubes were spun for 15 min at 12 k rpm at 4°C. The aqueous phase was carefully retrieved and transferred to a fresh tube and equal volumes of absolute ethanol was added. For better precipitation of RNA, 1 µl of glycogen (Roche, cat# 10901393001) was added to the tubes and incubated at −20°C overnight. Later, the tubes were spun down at 12 K rpm for 15 min at 4°C and the resultant pellet was washed in 70% ethanol, air dried, and subjected to DNase treatment using the manufacturer's instructions of Turbo DNA free (cat# AM1907). The DNA free RNA was further purified using Zymo Direct-zol RNA Micro-Prep kit (cat# R2050). The resultant pure RNA was reverse transcribed using Bio-Rad iScript Select cDNA synthesis kit (cat# 1708896) and SyBr green (cat# 1708880) based qRT-PCR was performed to determine the expression levels of AMP genes. For total RNA isolation from whole larvae, the above protocol was followed except no overnight precipitation method was used and that the larvae were homogenized using RNase-free pestles. qRT-PCR primers were designed using FlyPrimerBank (*Hu et al., 2013*).

## Statistics

All statistics with regard to the intensity measurements, cell counts, and qRT-PCR, were performed on Prism eight software. The error bars represent ± standard error of mean (SEM) or standard deviation (SD) as mentioned in the results section or figure legends. Significance between two conditions or genotypes was calculated by unpaired t test, while those involving multiple genotypes and conditions were calculated by one-way ANOVA on Prism 8. The p values shown in the figures are represented by * ($p<0.05$), ** ($p<0.01$), *** ($p<0.001$), and **** ($p<0.0001$).

## Acknowledgements

We thank Dr. M Chatterjee, Dr. S Boswell, and A Ratner of the single-cell core facility for their kind support in the inDrops-based cell encapsulation. We thank Drs. K Brückner, SE Mohr, J Zirin, M Arris, B Ewen-Campen, R Vishwanatha, R Rajakumar, A Ghosh, P Saavedra, L Liu, R Binari and all members of the Perrimon Lab for their critical comments and helpful insights on the manuscript. We thank Dr. P Montero Llopis and R Stephansky of the Microscopy Resources on the North Quad (MicRoN) core facility for their help in imaging, and the *Drosophila* RNAi Screening Center (DRSC) and Bloomington *Drosophila* Stock Center (BDSC) for providing fly RNAi and GAL4 lines used in this study. We also thank Dr. I Andó for the generous gift of the hemocyte specific antibodies.

NP is an investigator of the Howard Hughes Medical Institute. JS is an investigator of the Samsung Science and Technology Foundation under Project Number SSTF-BA1701-15. Work by VB, MS, and SHS at the Harvard Chan Bioinformatics Core was funded by the Harvard Medical School Tools and Technology Committee and with the support of Harvard Catalyst, The Harvard Clinical and Translational Science Center (NIH award #UL1 RR 025758 and financial contributions from participating institutions). The content is solely the responsibility of the authors and does not necessarily represent the official views of the National Center for Research Resources or the National Institutes of

Health. Portions of this research were conducted on the Orchestra High Performance Compute Cluster at Harvard Medical School. This NIH-supported shared facility is partially provided through grant NCRR 1S10RR028832-01.

## Additional information

### Competing interests

Jiwon Shim: Reviewing editor, *eLife*. The other authors declare that no competing interests exist.

### Funding

| Funder | Grant reference number | Author |
|---|---|---|
| Samsung Science and Technology Foundation | SSTF-BA1701-15 | Jiwon Shim |
| Howard Hughes Medical Institute | | Norbert Perrimon |
| Harvard Medical School | | Victor Barrera<br>Michael J Steinbaugh<br>Shannan Ho Sui |

The funders had no role in study design, data collection and interpretation, or the decision to submit the work for publication.

### Author contributions

Sudhir Gopal Tattikota, Conceptualization, Data curation, Formal analysis, Supervision, Validation, Investigation, Visualization, Methodology, Writing - original draft, Project administration, Writing - review and editing; Bumsik Cho, Formal analysis, Validation; Yifang Liu, Yanhui Hu, Victor Barrera, Michael J Steinbaugh, Sang-Ho Yoon, Fangge Li, Franz Dervis, Jin-Wu Nam, Shannan Ho Sui, Data curation, Formal analysis; Aram Comjean, Software; Ruei-Jiun Hung, Formal analysis; Jiwon Shim, Resources, Formal analysis, Validation, Writing - review and editing; Norbert Perrimon, Conceptualization, Resources, Supervision, Funding acquisition, Investigation, Visualization, Methodology, Writing - original draft, Project administration, Writing - review and editing

### Author ORCIDs

Sudhir Gopal Tattikota (iD) https://orcid.org/0000-0003-0318-5533
Victor Barrera (iD) http://orcid.org/0000-0003-0590-4634
Michael J Steinbaugh (iD) http://orcid.org/0000-0002-2403-2221
Sang-Ho Yoon (iD) http://orcid.org/0000-0003-2611-5554
Jiwon Shim (iD) http://orcid.org/0000-0003-2409-1130
Norbert Perrimon (iD) http://orcid.org/0000-0001-7542-472X

### Decision letter and Author response

Decision letter https://doi.org/10.7554/eLife.54818.sa1
Author response https://doi.org/10.7554/eLife.54818.sa2

## Additional files

### Supplementary files

• Supplementary file 1. Table representing number of cells, genes, reads, and unique molecular identifiers (UMIs) recovered per cell per sample.

• Supplementary file 2. Table representing the top marker genes per cluster pertaining to *Figure 1C and D*. One cluster per sheet.

• Supplementary file 3. Table representing the Differentially Expressed Genes per cluster across all conditions pertaining to *Figure 2* and its supplement.

• Supplementary file 4. Table representing differentially expressed genes across all conditions in PPO1low and PPO1high crystal cells.

• Supplementary file 5. Table representing differentially expressed genes across all conditions in lamellocyte clusters.

• Supplementary file 6. Table representing the gene enrichment analysis pertaining to *Figure 6A* and *Figure 3—figure supplement 2F*.

• Transparent reporting form

## Data availability

Sequencing data have been deposited in GEO under the accession number GSE146596. Elsewhere, data can be visualized at: www.flyrnai.org/scRNA/blood/. Data code can accessed at: https://github.com/hbc/A-single-cell-survey-of-Drosophila-blood (copy archived at https://github.com/elifesciences-publications/A-single-cell-survey-of-Drosophila-blood).

The following dataset was generated:

| Author(s) | Year | Dataset title | Dataset URL | Database and Identifier |
|---|---|---|---|---|
| Tattikota SG, Cho B, Liu Y, Hu Y, Barrera V, Steinbaugh MJ, Yoon S, Comjean A, Li F, Dervis F, Hung R, Nam J, Ho SS, Shim J, Perrimon N | 2020 | A single-cell survey of Drosophila blood | https://www.ncbi.nlm.nih.gov/geo/query/acc.cgi?acc=GSE146596 | NCBI Gene Expression Omnibus, GSE146596 |

The following previously published datasets were used:

| Author(s) | Year | Dataset title | Dataset URL | Database and Identifier |
|---|---|---|---|---|
| Miller M, Chen A, Gobert V, Augé B, Beau M, Burlet-Schiltz O, Haenlin M, Waltzer L | 2017 | Transcriptomic analysis of Drosophila larval crystal cells | https://www.ncbi.nlm.nih.gov/geo/query/acc.cgi?acc=GSE93823 | NCBI Gene Expression Omnibus, GSE93823 |

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
