## [Decision Letter]

**Acceptance summary:**

This article provides new insights on *Drosophila* larval hemocyte lineages using single cell RNA sequencing. This would be valuable for further characterization of the cellular immune response. It also reveals a new role of the FGF pathway in the encapsulation process with an interaction between crystal cells and lamellocytes.

**Decision letter after peer review:**

Thank you for submitting your article "A single-cell survey of *Drosophila* blood" for consideration by *eLife*. Your article has been reviewed by three peer reviewers, including Bruno Lemaitre as the Reviewing Editor and Reviewer #1, and the evaluation has been overseen by Anna Akhmanova as the Senior Editor.

The reviewers have discussed the reviews with one another and the Reviewing Editor has drafted this decision to help you prepare a revised submission. Please submit the revised version when you can – we are of course aware of the current difficult situation. As you will see, reviewers were quite positive about the manuscript. There are several comments that can be addressed with additional experiments. The hope is that you can strengthen your manuscript.

Summary:

This study by Tattikota et al. uses a single cell sequencing approach to investigate the heterogeneity of *Drosophila* blood cells and to characterize certain subpopulations across wounded, unwounded and parasitic wasp infected larvae. The study is an important addition to the literature and helps to demonstrate that even within these different blood cell types, hemocytes are not homogeneous. Another strong point was that the authors go beyond the simple characterization approach to single cell RNA seq and use the data to uncover an interesting requirement for FGF signaling in mediating immune responses against parasitoid wasp eggs, showing how, if followed up properly, scRNA seq data can lead to some very interesting biology. The paper also provides an extremely valuable resource for the insect blood cell community.

Essential revisions:

1) A recommendation is to better characterize the role of the FGF pathway. The finding should be backed up by the use of different RNAi and Gal4 lines. The phenotype 'a defect in wasp encapsulation' should be better characterized. What is the phenotype when the FGF ligand is over-expressed? Does FGF signaling play a role in melanization upon septic injury or in the process of crystal cell explosion? Are *bnl* and *btl* expressed also in LGs? With respect to wasp infection, PMs first attach to the wasp egg and then only will LM be recruited. It would have been interesting to determine whether *bnl* in PMs is required for this event. The authors ought to document the formation of the capsule around wasp eggs when they perturb FGF signaling and not simply rely on stainings performed on circulating hemocytes. It is also not clear whether these manipulations actually alter the immune response to wasps: is there any effect on the viability of larvae and the wasp hatching rate?

2) Whilst the authors discover potentially interesting subpopulations of hemocytes and attempt to classify them based on expression of key genes the study finds that essentially hemocytes can still be broadly classified into the three main types that have been known for many years, namely, plasmatocytes, crystal cells and lamellocytes. The question still remains whether the subpopulations identified by this study are permanent states or simply transient states that the individual cells have adopted temporarily. To investigate this thoroughly lies beyond the scope of the study but the authors should discuss this further.

3) Since the authors have delineated rather specific set of molecular markers or each hemocyte cluster, it would be an interesting addition to test those markers by RTqPCR to determine whether these different clusters are specific to circulating or sessile hemocytes.

---

## [Author Response]

Essential revisions:1) A recommendation is to better characterize the role of the FGF pathway. The finding should be backed up by the use of different RNAi and Gal4 lines.

With regard to the use of different GAL4 lines, we did use two independent GAL4 lines to knockdown *bnl*: *Hml-GAL4* (which is common to both plasmatocytes and most crystal cells) and *lz-GAL4* (specific to crystal cells), both of which yielded similar results. However, although we used two independent GAL4 lines (HLT- and srp-GAL4) to knockdown *btl*, only srp-GAL4 supported our findings of *bnl^RNAi^*. We believe that the *HLT-GAL*4 (*Hml-GAL4 UAS-FLP, ubi-FRT-STOP-FRT-GAL4*) may have inefficient flippase activity and/or lamellocytes may be contributed by non-Hml^+^ cells, thus resulting in a subtle phenotype. We also agree with the reviewer that additional RNAi lines are important to include. We used *UAS-bnl^RNAi^* (BL# 34572) as it has been widely used and validated (Du et al., 2018). In addition, we re-validated Figure 6—figure supplement 2A). With respect to *btl^RNAi^*, we now provide data with a second RNAi line against *btl* (BL# 60013) and demonstrate phenotypes that are similar to *srp-GAL4>btl^RNAi^-1* (please see updated Figure 6H; Figure 6—figure supplement 3A, C, D-I).this line by confirming its knockdown efficiency, which is ~45% (

The phenotype 'a defect in wasp encapsulation' should be better characterized. What is the phenotype when the FGF ligand is over-expressed?We appreciate the reviewer’s question about overexpressing the FGF ligand Bnl. In our experiments, knockdown of bnl resulted in defective melanization of wasp eggs, which is a simple readout. Overexpression experiments may provide interesting observations, however we believe that these are follow-up studies and beyond the scope of the current project, especially as the phenotype may not be straightforward to quantify and complicated to evaluate. In addition, our lab is currently shut down and it would take us months to perform these studies.Does FGF signaling play a role in melanization upon septic injury or in the process of crystal cell explosion?

We agree with the reviewer that testing the functions of FGF signaling will be interesting in the context of other modes of injury involving sepsis. While these experiments may help support our findings, we believe that these are follow-up studies and beyond the scope of the current project.

To address the question regarding the role of FGF signaling in crystal cell explosion, we believe that Bnl may play a role in maintaining crystal cell homeostasis as we observed a reduction in their numbers upon *bnl^RNAi^* in uninfested control condition. To address if this reduction is caused by enhanced crystal cell rupture, we imaged hemocytes derived from normal *lz-GAL4; UAS-EGFP>+* and *lz-GAL4; UAS-EGFP>bnl^RNAi^* larvae and found no evidence of crystal cell rupture (please see new Figure 6—figure supplement 2B-C). In separate experiments involving infestations in *Hml-GAL4>bnl^RNAi^* larvae, we imaged those few larvae (10%) that were able to melanize wasp eggs, because larvae that failed to melanize wasp eggs were technically challenging to stain and image. We observed intact (unruptured) crystal cells around wasp eggs in *Hml-GAL4-bnl^RNAi^* larvae compared to controls, which mostly did not show any crystal cells, presumably due to natural “rupturing” upon infestation (please see new Figure 6—figure supplement 2G-H; crystal cells are indicated by arrows). This observation suggests that Bnl may play a role in the process of rupturing, but further studies are warranted for comprehensively addressing its role in crystal cell homeostasis.

Are bnl and btl expressed also in LGs?

Yes, *bnl* and *btl* are indeed expressed in the lymph gland, albeit at lower levels in normal conditions, based on the lymph gland scRNA- and bulk RNA-seq comparative analysis. Furthermore, *bnl* and *btl* are enriched in crystal cell and lamellocyte clusters, respectively, in our lymph gland scRNA-seq data set available elsewhere (Cho et al., 2020). Please see Figure 6—figure supplement 1C-D for the data.

With respect to wasp infection, PMs first attach to the wasp egg and then only will LM be recruited. It would have been interesting to determine whether bnl in PMs is required for this event. The authors ought to document the formation of the capsule around wasp eggs when they perturb FGF signaling and not simply rely on stainings performed on circulating hemocytes.

We thank the reviewer for this question, especially when we noticed the expression of *bnl* in a small subset of plasmatocytes. We have now performed additional experiments to document the recruitment of Hml^+^ plasmatocytes in encapsulation of wasp eggs. We find that the number of plasmatocytes around the wasp eggs significantly reduced upon loss of *bnl* in plasmatocytes (please see new Figure 6—figure supplement 2G-I). This new data suggests that *bnl* may also be involved in regulating plasmatocyte recruitment towards wasp eggs. Further, based on our findings that Btl^+^ lamellocytes can be detected on wasp eggs (Figure 6E-E’’’), and that loss of *bnl* may affect the possible migration of plasmatocytes, we believe that FGF signaling may be important for capsule formation around wasp eggs.

It is also not clear whether these manipulations actually alter the immune response to wasps: is there any effect on the viability of larvae and the wasp hatching rate?

We did track the survival of *Drosophila* larvae post infestation in the *bnl^RNAi^* and *btl^RNAi^*. We found that the mortality rate of *lz-GAL4>bnl^RNAi^* larvae increased to 8.96% as compared to 4.71% in controls post wasp infestation, correlating with the reduced melanization seen upon *lz-GAL4>bnl^RNAi^*. However, we did not see a similar pattern of mortality rates in *Hml-GAL4>bnl^RNAi^* or *srp-GAL4>btl^RNAi^* wasp infested larvae.

2) Whilst the authors discover potentially interesting subpopulations of hemocytes and attempt to classify them based on expression of key genes the study finds that essentially hemocytes can still be broadly classified into the three main types that have been known for many years, namely, plasmatocytes, crystal cells and lamellocytes. The question still remains whether the subpopulations identified by this study are permanent states or simply transient states that the individual cells have adopted temporarily. To investigate this thoroughly lies beyond the scope of the study but the authors should discuss this further.

We thank the reviewer for this comment. Based on our pseudotime data, we believe that with exception to the activated PM states (e.g., PM^AMP^) and lamellocytes, most PM states may exist in transient states along the course of blood cell maturation in unwounded and wounded conditions. Due to the dynamic nature of *Drosophila* growth during larval stages, and that most plasmatocyte clusters exist during unwounded normal conditions, we suggest that certain PM states exist in a dynamic continuum along the course of blood cell maturation. For better clarity, we have now included some explanation with regards to the transient states in the Discussion section.

3) Since the authors have delineated rather specific set of molecular markers or each hemocyte cluster, it would be an interesting addition to test those markers by RTqPCR to determine whether these different clusters are specific to circulating or sessile hemocytes.

We thank the reviewer for this suggestion. We performed quantitative real time PCR on RNA derived from sessile and circulating hemocytes in unwounded condition. We conclude that most plasmatocyte clusters are equally represented in both compartments based on the expression of certain marker genes obtained from our scRNA-seq data. However, the PM clusters expressing *Ubx* and *mthl7* may be relatively more enriched in sessile compartments (please see Figure 1—figure supplement 1G).